# MemReasoner: A Memory-augmented LLM Architecture for Multi-hop Reasoning

## Abstract

Recent benchmarks suggest that there remains significant room to improve large language models' ability to robustly reason across facts distributed in extremely long documents. In this work, we propose MemReasoner, a new memory-augmented LLM architecture that is equipped to perform temporal reasoning, along with multiple computational steps, over the context stored in the latent memory, and is trained with supervision on intermediate steps and final outcome. Experiments show that MemReasoner trained on the core reasoning facts generalizes better, compared to the off-the-shelf large language models as well as fine-tuned recurrent models, on an unseen test distribution where the required facts are scattered across long natural text up to 128k tokens. Further, MemReasoner demonstrates robust reasoning performance relative to the baselines, when the answer distribution or number of hops in test samples differs from that in the training set.

## 1 Introduction

Transformer-based large language models (LLMs) have recently shown impressive performance in many natural language processing (NLP) tasks, including machine translation, question answering, and reading comprehension, demonstrating signature of general reasoning abilities. However, when restricted to individual NLP reasoning benchmarks, particularly those that require logical reasoning, current LLMs typically perform poorly despite efforts to improve accuracy through prompt engineering (Wei et al., 2022; Min et al., 2022). As such, more evidence seems to support the hypothesis that powerful LLMs often learn statistical features and correlations to simulate reasoning rather than performing true reasoning (Ruder, 2021).

The recently introduced BABILong benchmark further establishes this point, as it is designed to test LLM's ability to reason across facts distributed in extremely long documents (Kuratov et al., 2024). BABILong is developed based on the bAbi benchmark (Weston et al., 2015), which is composed of 20 reasoning tasks. These include fact chaining, simple induction, deduction, counting, and handling lists/sets (Weston et al., 2015). This set of tasks was designed as prerequisites for any system that aims to having a conversation with a human. BABILong further introduces irrelevant natural text from the PG19 book corpus (Rae et al., 2019) into the original context to make it artificially longer and include distracting text, while the underlying reasoning task remains the same. For examples of the task samples in BABILong, see Figure 1. Experiments with popular transformer-based LLMs shows that present days' transformer-based language models effectively utilize only 10-20% of the context and their performance declines sharply with increased reasoning complexity. Retrieval-augmented generation with LLMs at best can provide 60% accuracy for a simple QA task that requires extracting single evidence from the context. Interestingly, a memory-augmented transformer architecture, namely Recurrent Memory Transformers (RMT) (Bulatov et al., 2022) shows the highest performance on BABILong benchmark; suggesting that the long-term recurrent memory of the context helps. RMT in that case is trained on longer BABILong samples with supervision on final answer. However, as we will demonstrate in this work, RMT when trained on bAbi samples with supervision on final answer reconstruction, does not generalize well on BABILong test set. This observation suggests that memory-augmented LLMs can further benefit from additional supervision, when available. Additionally, enabling multiple hops over the memory per final answer reconstruction can help the model perform well, when the task demands so.

In this work, we provide an alternative language model architecture that is designed to naturally handle recurrent processing over long context *that is not seen by the model during training*. Our goal is to provide a more effective and robust solution for handling multi-hop generative QA tasks,

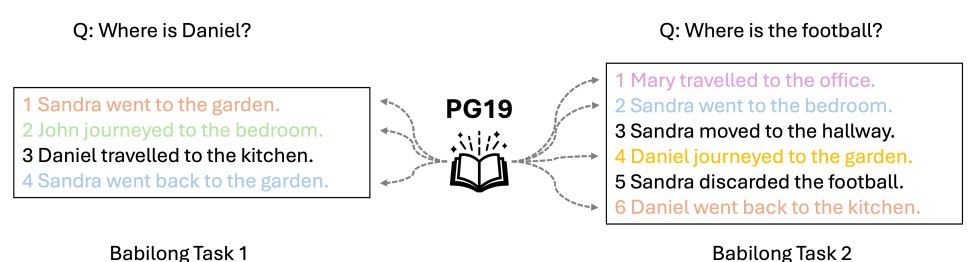

Figure 1: Examples of BABILong tasks.

which require the model to gather, relate, and reason over disjoint pieces of information from the *unseen* long context to generate an answer. Towards this goal, we propose a memory-augmented LLM architecture enhanced with two basic operations: (i) explicit learning of temporal orders of facts/events present within the context, and (ii) mechanism for iteratively reading from the context and updating the query accordingly. We refer to this new architecture as MemReasoner.

The backbone memory-augmented LLM used in this study is Larimar (Das et al., 2024), which is trained such that the latent encodings of a set of facts, referred as an episode, are written to a memory module. For a given query, the readout from this episodic memory module conditions the generation of the decoder, which is achieved by learning a differentiated attention to the readout during training. During inference, memory is dynamically updated by solving a linear system of equations, which is efficiently done via computing matrix pseudoinverse rather than gradient backpropagation. The memory mechanisms in Larimar assume order invariance of samples within an episode and support only single time read over the episode, which are insufficient for the architecture to handle more complicated tasks like multi-hop question-answering (QA). We note that our approach could in principle be used in conjunction with other LLMs augmented with an (episodic) memory module beyond Larimar.

Here, we extend the basic episodic memory module to act as a reasoning module by introducing a recurrent network, such as a GRU, which is tasked to capture the sequence of events/facts in the context written to the memory. This step prepares the inputs to the reasoning module with a structured understanding of their temporal relationships, which is critical for reasoning over time-varying information. For example, in the sample shown in Fig 1 (right), understanding that "Sandra moved to the hallway" happens before "Sandra discarded the football" is crucial to answer the question "Where is the football?" (Answer: hallway). Around the reasoning module, we further enable iterative reads from the memory to "hop" between supporting facts and update the query accordingly. This operation allows the model to dynamically retrieve and refine information across multiple computational steps performed over the context episode. These two operations around the latent memory allows in-depth deliberation over the context, which is then used by the decoder for generation. Our main contributions are:

- A novel memory-augmented LLM architecture, namely MemReasoner, which is equipped with temporal processing and iterative read over the context written to an episodic memory module and is trained with supervision on both intermediate and final reasoning steps.

- Evaluation of MemReasoner on the single-hop (Task 1) and two-hop (Task 2) QA tasks (see Figure 1) using the challenging BABILong benchmark, establishing that the proposed architecture can generalize to long context that is unseen during training, whereas off-the-shelf vanilla transformer-based LLMs struggle and alternative recurrent models trained to output final answer fail to generalize.

- Experiments showing that the proposed MemReasoner architecture indeed learns multi-step processing over the context to solve the QA task, as evident by its robust performance when the answers in the training data differ from those in the test samples within the same task, and when the model trained on bABi task 2 is tested on longer BABILong task 1 samples.

## 2 RELATED WORK

**LLM Reasoning** Logical reasoning, a critical aspect for advancing many scientific fields, involves deducing new conclusions from existing facts and rules. To derive the final answer, such reasoning challenges often require multiple steps to be executed effectively and in the right order. For instance, with facts like "John picked up the football" and "John went to the bedroom", a logical process will be to deduce that the football's current location is bedroom. Despite showing advanced ability to learn from instructions and in-context demonstrations to answer questions (Brown et al., 2020; Min et al., 2022), LLMs struggle with complex logical reasoning, especially multi-step reasoning (Liu et al., 2023a). This failure has been attributed to the autoregressive nature of LLMs (Stechly et al., 2024), which can be characterized by "System 1" (Kahneman, 2011), a mode of thought that is fast, instinctive but less accurate. To address this limitation, recent work proposes prompting LLMs to mimic generating intermediate chain of thought (CoT) reasoning steps (Wei et al., 2022), providing access to external tools/verifiers (Schick et al., 2023), or a combination of both (Paranjape et al., 2023), to mimic the process of generating deliberative and logical thinking steps, i.e., the "System 2" mode. Another direction currently being explored is to train reward models to rank the candidate solutions or rank the intermediate steps (Khalifa et al., 2023; Wang et al., 2024). Different from these works, MemReasoner does not rely on deliberate prompt engineering or access to external tools, neither does it require feedback from an external reward model. Instead, inspired by the distinction between System 1 and System 2-like thinking, MemReasoner utilizes the decoder for fast generation and the memory module for slow reasoning, which are two components tightly integrated via training. In that sense, MemReasoner is closer to the line of works that use (generated) rationales for supervised finetuning or for preference tuning of LLMs to enhance their reasoning abilities (Zelikman et al., 2022; Pang et al., 2024). However, it remains unexplored how those approaches perform on iterative reasoning tasks over lengthy context that is unseen during training.

**Long-context Modeling** The scope of the present study encompasses two distinct challenges around multi-step reasoning tasks, namely (1) processing very long context and (2) "hopping" over that context in a temporally-aware manner to link disjoint pieces of information and generate answers based on that. On the first challenge, vanilla transformer-based models struggle due to quadratic time and space complexity of self-attention and the increasing memory requirement of the key-value cache during generation. Recently, there has been significant progress in long-context modeling with transformers by using a mix of local and global attention (Munkhdalai et al., 2024), by continued pretraining on longer sequences (Xiong et al., 2023; Ding et al., 2024), by context window sliding and segmentation (Ratner et al., 2023), and by applying position extrapolation or interpolation to extend input length beyond the training phase (Press et al., 2022; Su et al., 2023). Promising alternative directions include the development of novel recurrent architectures (Bulatov et al., 2022) and state-space-models (Gu & Dao, 2023). Nevertheless, many of these techniques require training on longer sequences. Additionally, a number of studies and benchmarks suggest that the long-context LLMs may not be able to fully utilize their context window, and therefore performance degrades on simple retrieval and complicated reasoning tasks as the input length grows and/or the position of the answer varies within the context (Hsieh et al., 2024; Yuan et al., 2024; Liu et al., 2023b; Levy et al., 2024).

**Status Check on LLM Reasoning** Consequently, in parallel to impressive advances in LLMs abilities, caution has been raised on the discrepancy between claimed reasoning abilities as per standardized benchmarks and true reasoning skills. The scientific community has advocated for careful investigations of issues such as data contamination, performance robustness and generalization, and flawed reasoning benchmark that supports "shortcut learning" (Mitchell, 2023; Wu et al., 2024). For example, the presence of reasoning shortcuts in the task samples themselves has been reported in the HotPotQA dataset, which does not ensure language models are actually being required to perform multi-hop processing over the context (Jiang & Bansal, 2019). Recently, a number of tasks and benchmarks have been developed to address these issues (Valmeekam et al., 2022; Kuratov et al., 2024; Nezhurina et al., 2024). Along this line, we here show the generalization robustness of MemReasoner across (i) "unseen" context that consists of varying length of irrelevant natural text and (ii) answer distribution that is different from the training distribution.

## 3 MULTI-STEP REASONING WITH MEMREASONER

The key components of MemReasoner involve an LM encoder, an episodic memory module, and an LM decoder (see Figure 2a). The role of the episodic memory module is to enable *write* of the

context encodings in the memory, to allow performing search over the context encodings and *read* from them, in order to feed the decoder to execute the task. Given a logical reasoning task for which the supporting facts (reasoning process) and the final answer (reasoning outcome) are available, the MemReasoner architecture is trained to recover the supporting facts and the final answer. A search in the latent memory space is performed during training in order to correctly output the final answer and the supporting facts. An additional point worth mentioning is that, MemReasoner also is trained to learn the relative order of the supporting facts in the context, which is crucial for reconstructing an agent's or object's most recent location, as required by the bAbi reasoning tasks. Details are provided below.

### 3.1 PRELIMINARY

Let $\mathcal{X}$ be the LM input space, $\mathcal{Z}$ be the latent space, and $\mathcal{Y}$ be the LM output space. Larimar (Das et al., 2024) features an encoder $e$ that maps an input to an embedding $z \in \mathcal{Z} \subseteq \mathbb{R}^D$, and a memory module $\mathcal{M}$. The memory $M$ is adaptable in the sense, that it allows "write" and "read" operations as episodes (aka, contexts $C$, where each context is comprised of $E$ sentences) arrive, i.e., $\hat{M} = write(M, z), z_{\text{read}} = read(\hat{M}, z)$, wherein $\hat{M}$ is the updated memory after an write. And, a decoder $d$ that performs generations conditioned on the memory readout $z_{\text{read}}$.

### 3.2 LARIMAR FRAMEWORK

Now, suppose one is given an input context $C = \{c_1, ..., c_E\}$ with $E$ denoting the length of the context, and the target task is to answer a question $q$ conditioned on the given context $C$. To approach the task within the Larimar framework, the input, both context $C$ and query $q$, are encoded to their latents ($z_1, \ldots, z_E$ and $z_q$) via the encoder $e$. Next, let $M_0$ be the initial memory, write the context to the memory via a $write$ operation. To do so, Larimar follows the earlier works on Kanerva Machine (Wu et al., 2018), which is inspired by Kanerva's sparse distributed memory model (Kanerva, 1988), where the memory is viewed as a global latent variable in a generative model. In this framework, the goal is to learn a memory dependent data prior and learnable addresses, where the memory update and read/write are considered as Bayesian inference, i.e., the posterior parameters are updated as new data arrives. Later, (Pham et al., 2022) reformulated the encoding of new memories and decoding data from memories from Bayesian updates to an equivalent minimization problem, which essentially amounts to solving a linear system of equations, efficiently done via computing matrix pseudo inverses indicated by † hereafter. Therefore, memory is updated via the *write* operation such that, $\hat{M} = (Z_\xi M_0^\dagger)^\dagger Z_\xi$, where $Z_\xi = [z_1 + \xi_1, z_2 + \xi_2, \ldots, z_E + \xi_E]$ and $\xi_i \sim \mathcal{N}(0, \sigma_\xi^2 I)$.

Then, the $read$ operation translates the query embedding from the lens of the encoded memory to a query readout $z_r$ via $z_r = (z_q \hat{M}^\dagger + \eta)\hat{M}$, where $\eta \sim \mathcal{N}(0, \sigma_\eta^2 I)$. Lastly, the decoder $d$ decodes the query $q$ conditioned on the readout by using a *learnable* broadcasting parameter $W_M$ that casts $z_r$ to each decoder layer and obtains $h_k^m$ that serves as the past key values for $k = 1, \ldots, L$, where $L$ is the number of layers in the decoder.

We use this memory-augmented LLM architecture of Larimar and the operations as backbone for MemReasoner, due to its memory and space-efficient read/write abilities and demonstrated generalizability at test-time. It is worth mentioning the earlier works on memory-augmented neural nets, which use a recurrent neural net together with an external memory, have investigated ideas like temporal feature learning and iterative hops over context, for example, see (Weston et al., 2014; Sukhbaatar et al., 2015). However, to our knowledge, this is the first study to enable those operations around the explicit episodic memory of a transformer-based LLM during training and test the resulting model's generalizability on a long-context reasoning benchmark like BABILong.

### 3.3 MEMORY WITH TEMPORAL ORDER

Recall, the latent encoding of facts $\{z_1, ..., z_E\}$ within a context episode $C$ are written in the memory $M$ in an order-invariant manner. However, many multi-step reasoning tasks require some notion of temporal context. For example, when answering "where is John?" in the context of "... John is in the bathroom. ... John goes to the garden." ("..." denotes irrelevant facts), there should be a mechanism in place to guarantee that the memory encodes the correct temporal order of the facts, and the readout should reflect "John goes to the garden." as the supporting fact instead of "John is in the bathroom.".

To introduce some temporal notion within the context, in MemReasoner we introduce a temporal encoding module $\mathcal{P}$ that transforms *un-ordered* fact latents $\{z_1, ..., z_E\}$ within a context episode to

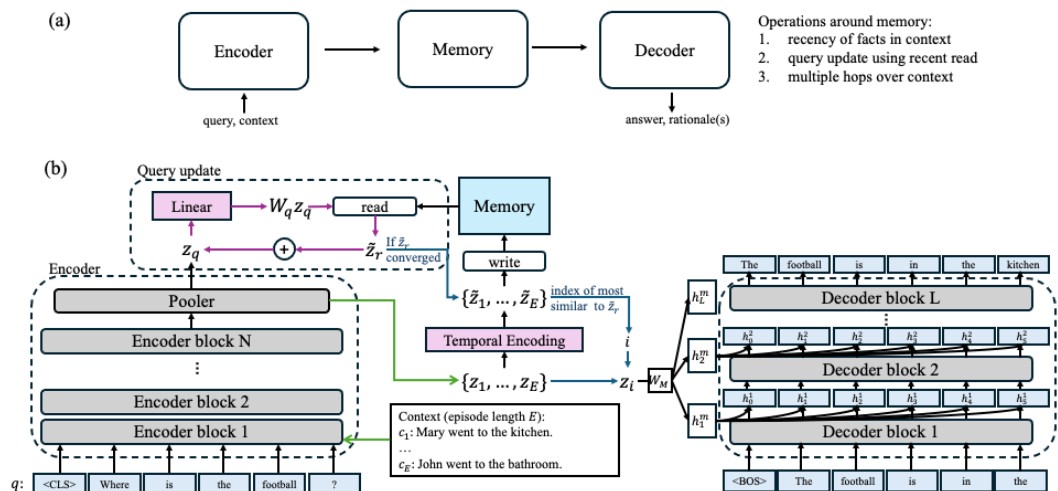

Figure 2: A diagram of the pipeline for reasoning with MemReasoner. (a) Conceptual overview of the framework. (b) Detailed architecture. $q$ denotes the query, $c_1, ..., c_E$ denotes the context for answering the query. $z_q$ denotes the encoding of the query while $\{z_1, ..., z_E\}$ denote encodings of each line of the context. We use $\tilde{z}$ to denote temporally encoded latents.

their *ordered* counterparts $\{\tilde{z}_1, ..., \tilde{z}_E\}$. The temporal encoding module is generic and allows any structure featuring sequentiality within context. In practice, we investigate two general types of encoding methods, *un-parameterized* methods such as Sinusoidal Positional Encoding and *parameterized* methods such as GRUs.

**Positional Encoding.** We compute positional encodings for each line of context within the episode by using sine and cosine functions similar to (Vaswani et al., 2017). Additionally, we experiment with positional encoding which assigns encodings starting from the last element of the episode. The structure ensures that for contexts of different length, the last lines of the contexts are encoded similarly, which is useful for QA tasks in which the most recent information is more relevant for answering the question.

Finally, to convert $\{z_1, ..., z_E\}$ to $\{\tilde{z}_1, ..., \tilde{z}_E\}$ with positional encodings, we add the computed positional encodings to the input.

*GRU.* We also investigate learnable encodings via a bidirectional GRU unit. For these, we treat $\{z_1, ..., z_E\}$ as the sequence passed as input into the GRU and simply let $\{\tilde{z}_1, ..., \tilde{z}_E\}$ be the sequential outputs of the GRU.

These ordered context embeddings $\{\tilde{z}_1, ..., \tilde{z}_E\}$ are then written to memory via Larimar's *write* operation.

### 3.4 Iterative Read And Query Update

A typical multi-step reasoning task often inherently requires "hops" between facts until the final solution is found. Additionally, the query embedding can be updated accordingly to reflect the most recent hop.

In order to perform hopping between facts, we first recall the three key components interacting with the memory module $\mathcal{M}$, the fact embeddings ($\{z_1, ..., z_E\}$) within a context episode, the query embedding $z_q$, and the memory readout $z_r$. Let us further consider M stores facts that have been ordered temporally $\{\tilde{z}_1, ..., \tilde{z}_E\}$.

To enable *iterative read*, we pass $z_q$ through a linear layer to obtain $\hat{z}_q = W_q z_q$ before the *read* operation from the memory, where $W_q \in \mathbb{R}^{D \times D}$ is a learnable parameter that absorbs the scale changes introduced by the position encoding in the memory. Specifically, different from Section 3.1, here we have $z_r = (\hat{z}_q \hat{M}^\dagger + \eta)\hat{M}$.

To *update the query*, we first update the query latent and let $z_q \leftarrow z_q + \alpha \cdot z_r$, where $\alpha \in \mathbb{R}$ is a hyperparameter to balance the load from the previous readout. The updated query is then fed into the memory module for another $read$ operation to obtain a new $\tilde{z}_r$. The query update procedure is repeated until the readout converges (i.e. $||\tilde{z}_r^t - \tilde{z}_r^{t+1}||_2 < \tau$ where $\tilde{z}_r^t$ denotes the readout at time $t$ and $\tau$ is a hyperparameter) or until it reaches a fixed number of maximum iterations.

## 3.5 FULL WORKFLOW

Now that we have discussed all components of MemReasoner, we elaborate the full pipeline in the following and provide a visualization in Figure 2b.

Consider an input context $C = \{c_1, ..., c_E\}$, a question $q$, an encoder $e$, a temporal encoding module $\mathcal{P}$, an initial memory module $\mathcal{M}$, and a decoder $d$. We first encode the context $C$ and query $q$ to their latents, $z_1, \ldots, z_E$ and $z_q$, via encoder $e$. Then, we follow Section 3.3 and transform $z_1, \ldots, z_E$ to $\tilde{z}_1, ..., \tilde{z}_E$. Next, we write the ordered context $\tilde{z}_1, ..., \tilde{z}_E$ to the memory and obtain $\hat{M}$. Subsequently, we read using the query latent from the memory and perform query and read updates according to Section 3.4. After we have obtained a $\tilde{z}_r$ as a final readout which does not undergo update anymore, we map $\tilde{z}_r$ to the corresponding unordered encoding in $M$. This is because we only want the additional position information to be used when locating the most relevant contexts, but not during the decoding - if being fed to the decoder, the decoder may overfit to the ordering information in the latents. We do this by first finding the index of the most similar ordered latent encoding $i = \arg\min_{j \in \{1, ..., E\}} ||\tilde{z}_r - \tilde{z}_j||_2$ and then obtaining the corresponding encoding $z_i$ from the unordered encodings (prior to undergoing temporal encoding in Figure 2) $\{z_1...z_E\}$. Lastly, the decoder $d$ decodes the prompt $P_a$ given for answer generation conditioned on $z_i$. We provide the full pseudocode in Algorithm 1.

## 3.6 TRAINING OBJECTIVES

Let $\mathcal{D}_{\text{reason}}$ denote the reasoning data distribution while $\mathcal{D}_{\text{pretrain}}$ denotes the pretraining data distribution. Each sample from $\mathcal{D}_{\text{reason}}$ is of the form $(q, C, S, a)$ where $q$ is the query, $C = \{c_1, ..., c_E\}$ are the facts in the context, $S$ is a set of indices corresponding to supporting facts (we will use $S_i$ to denote the $i$th supporting fact index in $S$), and $a$ is the answer. Meanwhile the pretraining distribution corresponds to a generic corpus, e.g. Wikipedia. Let $e$ denote the encoder, $d$ denote the decoder, $t$ denote temporal encoding, $\tilde{z}_r^i$ denote the $i$th temporally encoded readout from iterative reading with $\tilde{z}_r^0 = q$, $z_r^i$ represent the unordered encoding corresponding to the $i$th ordered readout, and $P_a$ and $P_s$ denote the prompts for generating the answer and supporting fact respectively. To train the model, we utilize the following loss function in Equation 1.

$$L = \mathbb{E}_{(q,C,S,a) \sim \mathcal{D}_{\text{finetune}}} \left[ \underbrace{\mathbb{E}_{z_r^{|S|} \sim p(z_r^{|S|}|q,M,\tilde{z}_r^0...\tilde{z}_r^{|S|-1})} \ln p(a|z_r^{|S|}, P_a)}_{\text{reconstruction of answer}} \right.$$

$$+ \alpha \sum_{i=1}^{|S|} \underbrace{\mathbb{E}_{z_r^i \sim p(z_r^i|M,\tilde{z}_r^0...\tilde{z}_r^{i-1})} \ln p(c_{S_i}|z_r^i, P_s)}_{\text{reconstruction of supporting facts}} + \beta \sum_{s \in S} \underbrace{\ln p(d(e(c_s)))}_{\text{autoencoding of supporting fact}}$$

$$\left. + \delta \sum_{i=1}^{|S|} \underbrace{\mathbb{E}_{\tilde{z}_r^i \sim p(\tilde{z}_r^i|q,M,\tilde{z}_r^0...\tilde{z}_r^{i-1})} \ell_{\text{order}}(\tilde{z}_r^i, S_i)}_{\text{ordering loss}} \right] + \rho \underbrace{\mathbb{E}_{x \sim \mathcal{D}_{\text{pretrain}}} \ln p(d(e(x)))}_{\text{autoencoding of pretraining dataset}} \quad (1)$$

$\alpha, \beta, \delta$ and $\rho$ are hyperparameters controlling regularization strength and $\ell_{\text{order}}$ is given by

$$v(z_r) = \text{softmax}([-||t(e(c_1)) - z_r||_2, ..., -||t(e(c_E)) - z_r||_2]^\mathsf{T})$$
$$\ell_{\text{order}}(z_r, s) = -\ln v(z_r)_s \quad (2)$$

The first and second terms correspond to the reconstruction loss of the answer and the supporting fact(s) with respect to the corresponding prompt for obtaining the answer $P_a$ and final readout, the third and the fifth terms correspond to the autoencoding loss of the supporting fact(s) and pretraining

---

**Algorithm 1:**

```
1  ]  Function IterativeRead( q, {c_1,...,c_E}, α, τ ):
       // q denotes the query tokens while {c_1,...,c_E} denote the E
          lines of context tokens, α is a hyperparameter for the
          query update, τ is a threshold hyperparameter for
          terminating iterations, P_a is the prompt given to the
          decoder for answer generation
       // encode query and context lines with encoder
2      z_q ← encode(q) for i ← 1 to E do
3      |   z_i ← encode(c_i)
4      end
       // apply temporal encoding over the sequence of context lines
          and write to memory
5      z̃_1,...,z̃_E ← temporalEncoding(z_1,...,z_E)
6      M̂ ← write(z̃_1,...,z̃_E)
       // iterative read and query update
7      z̃_r ← queryUpdate(z_q, α, τ)
       // Map to latent prior to performing temporal encoding
8      i* ← arg min_{i∈{1...E}} ||ẑ_i − ẑ_r||_2
9      return decode(z_{i*}, W_M, P_a)            // generate the answer with the
        decoder, W_M is a learnable parameter which interfaces the
        z_{i*} with the decoder
10
11
12 Function temporalEncoding({z_1,...,z_E}, method):
       // temporally encode the sequence {z_1,...,z_E}
13     if method = PE then
14     |   return {z_i + PE(i)| ∀i ∈ {1,...,E}}
15     else if method = GRU then
16     |   return GRU({z_1,...,z_E})
17
18 Function queryUpdate(z_q, α, τ):
       // given the query encoding z_q and threshold τ, perform
          iterative reading and update query
19     z̃_r ← read(W_q z_q, M)                       // W_q is learned parameter
20     z_q = z_q + α z̃_r                            // query update
21     z̃_{r,next} ← read(W_q z_q, M)
22     do
23     |   z̃_r ← z̃_{r,next}
24     |   z_q = z_q + γ z̃_r
25     |   z̃_{r,next} ← read(W_q z_q, M)
26     while ||z̃_{r,next} − z̃_r||_2 > τ
27     return z̃_r
28
```

---

data. The fourth term is a loss for encouraging the index of the most similar entry (by l2 distance) to the ordered readout at each iteration to match the index of the supporting fact through computing the cross entropy.

## 4 EXPERIMENTAL DETAILS AND RESULTS

### 4.1 DATASETS AND DATA PRE-PROCESSING

In the main paper, we utilize tasks 1 and 2 from the synthetic bAbi benchmark as our testbed. We also report results on Variable Tracking task from the RULER benchmark (Hsieh et al., 2024) in appendix. The bAbi datasets were prepared by synthesizing relations among characters and objects across various locations, each represented as a fact, such as "Mary traveled to the garden". Task 1

requires performing a single hop to find answer, whereas task 2 requires gathering two supporting facts in the right order (see Fig 1). These single to multi-hop QA tasks from BABILong benchmark together provide a controlled setting for evaluating LLMs' ability to reason over long context, where the difficulty of the task can be varied by changing the length of irrelevant text. The nature of this benchmark, where the synthetic sentences corresponding to the actual reasoning task are hidden inside irrelevant but lengthy naturally occurring text, keeps it at a low risk of data contamination to training sets of todays' LLMs. And finally, BABILong leaderboard shows tasks 1 and 2, while being simple enough, are challenging enough for off-the-shelf LLMs to solve.

We finetune MemReasoner separately on original bAbi task 1 and task 2 training split, each consisting of 10k samples (Weston et al., 2015). We then evaluate on the test set of the corresponding task from bAbi as well as from BABILong (Kuratov et al., 2024), in which the core reasoning facts from bAbi is distributed over arbitrarily long documents. Here we benchmark MemReasoner on BABILong test samples of up to 128k tokens.

For preprocessing bAbi data, we treat each training sample comprised of multiple facts as a single context episode, and individual sentence within that context as an instance within that episode. Each fact within an episode contains up to 64 tokens.For BABILong and for Wikipedia, if sentences are longer than 64 tokens, we split the sentences at multiples of 64 tokens.

We initiate MemReasoner finetuning from Larimar checkpoint pretrained on Wikitext (obtained by following the training protocol described in (Das et al., 2024)), which uses a Bert-large as the encoder and a GPT2-large as the decoder (For extension to MemReasoner with GPTJ-6B, see appendix). The number of parameters in MemReasoner is 1.4B. The slot size in the memory is 512. During finetuning, we randomly sample a batch of pretraining data (Wikipedia) of the same size as the batch of finetuning data (bAbi) for computing the autoencoding loss on the pretrain dataset of 2M samples. We generate the answer to the question by passing a prompt to the decoder (i.e. in the case of bAbi Task1-2, the prompt has the from "$<BOS>$ $X$ is in the" where $X$ denotes subject of the query).

We train MemReasoner models for 200 epochs using Adam optimizer with learning rate 5e-6. We set batch size to be 10. Additionally, we set query update parameter $\alpha = 1$. The maximum episode length varies from 14 (bAbi Task 1) to 72 (bAbi Task 2). Which means that MemReasoner has been exposed to a maximum of 90 and 573 tokens during finetuning on task 1 and task 2, respectively, whereas at test-time the model is exposed to contexts that are up to 128k tokens long. Since bAbi Task 1 is a single hop task, we do not perform query update during either training or inference. When fine-tuning on bAbi Task 2, we perform a fix number of 2 hop (equivalent to 1 query update) during the training. With bAbi Task 2 fine-tuned MemReasoner, we re-use the "2 hop" setting at inference on all tasks, including bAbi Task 2 and BABILong Task1/2. We consistently use query update parameter $\alpha$=8 throughout our experiments and include an ablation study on $\alpha$ in the appendix. Due to the page limit, we also defer ablation studies on the episodic memory, temporal encoding schemes, level of supervision, and the number of training epochs to the appendix.

## 4.2 BASELINE METHODS

**Off-the-shelf Baselines.** We show published results from (Yang et al., 2023) obtained using GPT-3 (175 B parameters) as an off-the-shelf baseline, with few-shot and chain-of-thought prompting, for comparison with MemReasoner on original bAbi test set. We also report performances of a recurrent memory transformer-0.77B and of a Mamba-1.4B model, which we fine-tune on bAbi samples, on bAbi test set. For BABILong benchmarking, we include the following models from BABILong leaderboard: (1) Meta-Llama-3-8B-Instruct with an 8K context window size, (2) Phi3-mini-128k-instruct – a long-context LLM consisting of 3.8B parameters and a 128k token long context window, and (3) Llama3-ChatQA-1.5-8B with a nvidia/dragon-multiturn-query-encoder – a RAG framework.

**Fine-tuned Baselines.** We add RMT-137M and Mamba-130M performances from BABILong leaderboard, which has been finetuned on a set of samples that belong to the same distribution as BABILong (with PG19 padding) but is not included in BABILong benchmarking test set. These models were finetuned by using a curriculum schedule that progressively increases sequence lengths: 1, 2, 4, 6, 8, 16 and 32 segments (Kuratov et al., 2024).

We further benchmark RMT and Mamba models finetuned on bAbi on BABILong test samples. The goal is to figure out if those alternative recurrent models perform well on BABILong leaderboard due

to their true learning ability of the underlying task or due to their exposure to BABILong samples during finetuning. We fine-tune off-the-shelf RMT(0.14b/0.77b) and Mamba (0.13b/1.4b) models using the next token prediction loss on final answer reconstruction on bAbi Task 1 and 2 separately till the testing accuracy on the task is sufficiently high (near 100%). In practice, we use 5 epochs to reach above 99% accuracy on RMT and 20 epochs for the accuracy to plateau on Mamba, all using Adam optimizer with learning rate $1e - 5$. RMT training was done with multiple segments using a curriculum learning procedure. In order to train with more segments while exposing the model to only bAbi data, we reduce the segment size to 64 for task 1 and 128 for task 2. This leads to 2 segments in training for task 1 and 2-4 segments in training for task 2. In order to mimic the curriculum learning process, we filter the data so that we train with inputs with token length up to the segment size for 10 epochs, up to 2 times segment size for another 10 epochs, and so on.

We also add a Larimar-1.3B baseline, which is finetuned on bAbi and Wikipedia samples with first and fifth terms from eqn. 1. The purpose of comparing MemReasoner with respect to Larimar is to disambiguate the benefits of temporal feature learning and iterative query and read updates on top of the episodic memory. Larimar fine-tuning shares the same training setups as MemReasoner. We further add experiments with Qwen2.5-0.5B (https://huggingface.co/Qwen/Qwen2.5-0.5B) and Qwen2.5-1.5B (https://huggingface.co/Qwen/Qwen2.5-1.5B) models (both of which support long context windows up to 128k tokens), as well as a memory network (Sukhbaatar et al., 2015) that is not coupled to transformer-based LLMs. See appendix for results.

It should be mentioned that all baselines used in this study are trained with supervision on final answer, whereas MemReasoner uses both supporting fact and final answer supervision. As mentioned earlier, the goal is to check if this additional supervision, when tied to the operations around the latent memory, enables better reasoning generalization. In that sense, MemReasoner offers a principled, model-agnostic approach for augmenting memory-based LLMs with robust reasoning, which can be complimentary or used together with continual training. Throughout the paper, we report task accuracy as the performance metric, so higher the better.

## 4.3 RESULTS

### 4.3.1 PERFORMANCE ON BABI TEST SET

Table 1 reports the performance of MemReasoner, which is independently finetuned on original bAbi task 1 and task 2, along with the baselines on the corresponding bAbi test set of 1k samples. Results show that, while prompting techniques such as few-shot learning and chain-of-thought prompting (Yang et al., 2023) work well on task 1 which requires a single hop to find the entity location, those baselines perform much poorly on task 2 that requires learning temporal dependence and performing multiple hops across facts to generate the final answer of object location. MemReasoner, as well as RMT, Mamba and Larimar, all finetuned on bAbi achieves near-perfect accuracy on both tasks. Importantly, Larimar baseline falls behind MemReasoner on both tasks, while the gap being bigger on more complicated task 2, implying that read/write to episodic memory alone is not sufficient.

### 4.3.2 PERFORMANCE ON BABILong TEST SET

Table 2 and Table 3 report accuracy of MemReasoner, together with baseline methods, on BABILong task 1 and task 2 samples, respectively. '-' means unavailable due to out of memory errors or maximal input length constraints. For task 1, the following observations can be made: (i) at half of model's context window, the accuracy of Llama-3-8B-Instruct drops to 80% and Phi-3-mini-128k drops to 63% of the corresponding model's performance at 0k samples, indicating LLMs are not good at utilizing their full context window. With RAG, the performance stays at a flat $\approx 60\%$ all throughout. Interestingly, while RMT and Mamba, when finetuned on BABILong samples of up to 16k tokens, are the best models reported on BABILong leaderboard, they perform poorly on BABILong samples beyond 0k as we finetune them on bAbi samples. This suggests exposure to BABILong during training helps RMT and Mamba, as the models have seen facts embedded inside the background distractor text from PG19. Larimar finetuned on bAbi, while performing much poorly on bAbi test set and BABILong 0k set to begin with, the accuracy on longer BABILong samples is higher than bAbi-tuned RMT and Mamba baselines. In contrast, MemReasoner trained on bAbi with supervision on supporting fact(s) and final answer generalizes well on BABILong for task 1, providing an average accuracy of 84.6% and 68.5% on $\leq$ 8k and $\geq$ 16k BABILong samples, respectively. These results suggest, that models can benefit on long-context reasoning from having access to longer similar sequences or to reasoning processes during training.

| Model type | Task 1 | Task 2 |
|---|---|---|
| CoT - GPT-3 | 97.3 | 72.2 |
| Few-shot - GPT-3 | 98.4 | 60.8 |
| RMT-.77B (bAbi) | 97.7 | 97.5 |
| Mamba-1.4B (bAbi) | **100** | 95 |
| Larimar-1.3B (bAbi) | 60.6 | 44.9 |
| MemReasoner-1.4B (bAbi) | **100** | **100** |

Table 1: Performance on bAbi tasks. Best model is highlighted in bold. GPT-3 (=text-davinci-003) baselines are from (Yang et al., 2023). Finetuning data, if any, seen by a model is specified within parentheses.

| Model type | Avg. $\leq$ 8k | Avg. $\geq$ 16k | 0k | 1k | 2k | 4k | 8k | 16k | 32k | 64k | 128k |
|---|---|---|---|---|---|---|---|---|---|---|---|
| RMT-.14B (BABILong)* | 100 | 97 | 100 | 100 | 100 | 100 | 100 | 100 | 99 | 96 | 94 |
| Mamba-.13B (BABILong)* | 100 | 100 | 100 | 100 | 100 | 100 | 100 | 100 | 100 | 100 | 100 |
| Few-shot - Meta-Llama-3-8B-Instruct* | 84.4 | - | 98 | **93** | **90** | **79** | 62 | - | - | - | - |
| Few-shot - Phi-3-mini-128k-instruct* | 78.4 | 38 | 97 | 84 | 72 | 69 | 70 | 60 | 53 | 38 | 1 |
| RAG - Llama3-ChatQA-1.5-8B* | 59.6 | 60 | 60 | 62 | 60 | 58 | 58 | 60 | 60 | 56 | 64 |
| RMT-.14B (bAbi) | 32.8 | 15.5 | 96 | 4 | 26 | 19 | 19 | 12 | 22 | 12 | 16 |
| RMT-.77B (bAbi) | 37.2 | 16.7 | 99 | 27 | 21 | 25 | 14 | 14 | 19 | 16 | 18 |
| Mamba-.13B (bAbi) | 20.4 | - | 85 | 11 | 5 | 0 | 1 | 0 | 0 | 0 | - |
| Mamba-1.4B (bAbi) | 44.2 | - | **100** | 60 | 42 | 19 | 0 | 0 | 0 | 0 | - |
| Larimar-1.3B (bAbi) | 44.8 | 14.3 | 63 | 59 | 55 | 28 | 19 | 14 | 16 | 13 | 14 |
| MemReasoner-1.4B (bAbi) | **84.6** | **68.5** | 99 | 91 | 83 | 76 | **74** | **71** | **68** | **70** | **65** |

Table 2: BABILong Task 1 Results. Baseline results marked with "*" are cited from (Kuratov et al., 2024). The finetuning data, if any, seen by each model is specified within parentheses.

| Model type | Avg. $\leq$ 8k | Avg. $\geq$ 16k | 0k | 1k | 2k | 4k | 8k | 16k | 32k | 64k | 128k |
|---|---|---|---|---|---|---|---|---|---|---|---|
| RMT-.14B (BABILong)* | 98.8 | 68.5 | 100 | 100 | 99 | 98 | 97 | 94 | 82 | 59 | 39 |
| Mamba-.13B (BABILong)* | 98.0 | 94.5 | 98 | 98 | 98 | 98 | 98 | 98 | 98 | 95 | 87 |
| Few-shot - Meta-Llama-3-8B-Instruct* | 40.2 | - | 47 | 46 | 49 | 39 | 20 | - | - | - | - |
| Few-shot - Phi-3-mini-128k-instruct* | 40.6 | 15.5 | 57 | 38 | 38 | 36 | **34** | **23** | **22** | 15 | 2 |
| RAG - Llama3-ChatQA-1.5-8B* | 21.6 | 8.75 | 28 | 25 | 22 | 19 | 14 | 13 | 9 | 7 | 6 |
| RMT-.14B (bAbi) | 36.6 | 12 | 97 | 31 | 19 | 16 | 20 | 12 | 12 | 14 | 10 |
| RMT-.77B (bAbi) | 41.2 | 17.5 | **100** | 36 | 21 | 27 | 22 | 18 | 23 | 13 | 16 |
| Mamba-.13B (bAbi) | 16.2 | - | 64 | 10 | 3 | 3 | 1 | 0 | 0 | 0 | - |
| Mamba-1.4B (bAbi) | 31.6 | - | 94 | 44 | 15 | 5 | 0 | 0 | 0 | 0 | - |
| Larimar-1.3B (bAbi) | 31 | **20.3** | 42 | 41 | 29 | 22 | 21 | 19 | 16 | **22** | **24** |
| MemReasoner-1.4B (bAbi) | **60.6** | 18.5 | **100** | **73** | **61** | **46** | 23 | 20 | 19 | 17 | 20 |

Table 3: BABILong Task 2 Results. Baseline results marked with "*" are cited from (Kuratov et al., 2024).

For more complicated task 2, which requires learning temporal dependence between the facts and finding and using two supporting facts in correct order for generation, both few-shot prompting and RAG with different base LLM show poor performance to begin with, and sharply degrade with context length increase of test samples. Again, RMT and Mamba, when fitted to BABILong, perform well on test samples, both struggle to generalize from bAbi to BABILong. For example, the accuracy drops from near 100% at 0k to 18% for RMT and to 36% for Mamba at 1k. Poor results at short context length for Larimar also indicates model's failure to learn the task. MemReasoner, in comparison, provides an accuracy of 100% at 0k, 73% at 1k, and 46% at 4k, while performance degrades to $\approx$ 18.5% beyond 16k. The modest ($\approx$ 18.5%) performance of bAbi-tuned MemReasoner at 16k or longer context suggests that there remains significant room for MemReasoner to improve, which will be investigated in future. One possible direction is to train MemReasoner on longer sequences and/or with different levels of supervision.

### 4.3.3 GENERALIZATION TO OUT-OF-DISTRIBUTION TEST SETS

To test if the models have indeed learned to solve the tasks, we create a new testbed where the construct of the tasks remains the same, but the answer changes from training to test set. Specifically, we change the location information present in the answer set of bAbi training $\rightarrow$ test as follows:

| Model type | Task 1 | Task 2 |
|---|---|---|
| RMT-.77B (bAbi) | 44.7 | 0.6 |
| Mamba-1.4B (bAbi) | 67 | 44 |
| Larimar-1.3B (bAbi) | 24.9 | 7.8 |
| MemReasoner-1.4B (bAbi) | **87.2** | 52.7 |

Table 4: Robustness to location changes in bAbi test set.

| Model type | 0k | 1k | 2k | 4k |
|---|---|---|---|---|
| RMT-.77B (bAbi) | **100** | 19 | 20 | 12 |
| Mamba-1.4B (bAbi) | 81 | 8 | 0 | 0 |
| Larimar-1.3B (bAbi) | 45 | 19 | 20 | 11 |
| MemReasoner-1.4B (bAbi) | 83 | 58 | **50** | **45** |

Table 5: Performance on bAbi task 2 → BABILong task 1 generalization.

office → library, garden→ garage, kitchen→ cafe, bathroom → attic, bedroom→ basement, hallway → gym. This now becomes a more stringent test, to which we subject all alternative architectures including MemReasoner. As shown in Table 4, RMT struggles in this setting across both tasks. On task 1, MemReasoner shows ≈ 20% higher accuracy than Mamba, whereas on task 2 MemReasoner wins by ≈ 8%.

Finally, we also check if the models trained on 2-hop bAbi task 2 can solve the simpler 1-hop task 1 but on the corresponding BABILong samples. Results are shown in Table 5, indicating that the best performing model on 0k BABILong task 1 samples is RMT, while MemReasoner being a second. However, both RMT and Mamba perform very poorly on longer (1-4k tokens) BABILong samples, whereas MemReasoner's accuracy remains strong.

## 4.4 CONCLUSION

In this work, we introduce a new memory-augmented LLM architecture that comes with two essential abilities required to perform robust multi-step reasoning, *i.e.*, learning temporal relations and to hop meaningfully between facts within a context. Our formulation and implementation of the multi-step reasoning mechanisms around the episodic memory, textcolortogether with supervised training using reasoning steps and final answer, is generic and in principle model-agnostic, and therefore can be leveraged to enhance other memory-augmented LLMs, including the ones used in this study as baselines. We examine MemReasoner on BABILong, a benchmark purposed to test models' reasoning ability when relevant facts are distributed in background of very large textual corpora. This deceptively lengthy nature of BABILong samples, along with the presence of distracting text that is naturally occurring, makes the underlying reasoning task more challenging on which even bigger LLMs that have seen samples with long context during training fails. We show here that, MemReasoner trained on bAbi samples provides strong performance on BABILong, compared to the off-the-shelf powerful LLM baselines and alternative recurrent architectures that are also fine-tuned on bAbi data, though only with final answer supervision.. We further show that MemReasoner generalizes better in the setting where answers in training set differs from those in the test within the same task. MemReasoner also shows good adaptation from two-hop to single-hop QA task, whereas the test samples are much longer and mixed with natural irrelevant text. Additional experiments show generality of MemReasoner approach across decoder scale (GPT2-l to GPTJ-6B) and across different multi-hop tasks. Results indicate, supervision on both supporting facts and final outcome, together with multi-hop search over the context in latent memory space, enables more robust reasoning generalization of LLMs. Taken together, designing alternative architectures with new loss objectives that encourage the model to learn the underlying reasoning skills is a potential path toward more robust reasoners.

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

# A APPENDIX

## A.1 WHY NOT FINETUNE ON LONGER SEQUENCES?

The goal of this work is to propose an LLM-based architecture that learns to reason over unseen long-context in an efficient, robust, and generalizable manner. As such, the evaluation framework corresponds to a set-up where the core reasoning facts are diluted in the presence of irrelevant natural text distractors distributed over the context. This setup allows one to test how consistently language model can solve the same reasoning task across different input lengths. This is inspired by the recent research showing that current LLMs' reasoning performance degrade at much shorter input lengths than their technical maximum (Levy et al., 2024). At the same time, finetuning on longer sequences presents several practical challenges: (i) The longer sequences with proper (human or machine) annotation should be available during training – which is typically expensive and is difficult to scale in real-world. (ii) Expanding the context window usually incurs a quadratic increase in computational and memory cost for transformer-based LLMs. For example, the training setup used in (Fu et al., 2024) shows that extending the Llama-2 7B model's context window from 4k to 80k requires 8 A100 GPUs (80G each) for five days. The costs of resources and time further increase significantly for larger models, for longer context length, and for more extended training period. (iii) The test distribution is still expected to match to longer sequences seen during training (e.g. mixture of bAbi with text from PG-19 in the case of BABILong) – which may not be always possible. As a result, straightforward continual pre-training or fine-tuning on longer sequences may still not fully solve the fundamental problem of learning to reason over (long) context in a robust and generalizable manner, and such approaches can benefit from using additional supervision (when available) and training-time search in the latent memory space.

While MemReasoner is not trained on longer samples that are similar to the test distribution (whereas RMT-BABILong and MAMBA-BABILong models in Tables 2 and 3 are), we train the model with additional supervision on supporting facts. In that sense, our MemReasoner approach with reasoning process and outcome supervision is complementary to the continual pre-training only with outcome supervision.

## A.2 ADDITIONAL DATASET PREPROCESSING DETAILS

In the unprocessed bAbi data, a single data instance consists of a sequence of lines representing facts to reason over with questions interspersed throughout the facts. We preprocess the bAbi data such that after pre-processing, a single training sample consists of a single question with facts for reasoning being the lines before it, with previous questions replaced by an empty line. On average, this leads to about 2 empty lines per training sample. For batches containing training samples with different lengths of context episodes, we pad shorter episodes with rows of the encoder padding token at the beginning.

## A.3 COMPARISON OF INFERENCE-TIME COMPLEXITY

Let $H_1$, $H_2$ and $d_1$, $d_2$ be the number of transformer layers and hidden state dimension in the encoder and decoder, respectively. Let $E$ denote the number of context lines in a sample, $L$ be the max context length, $L_1$ be the max query length, $D$ be the latent space dimension, and $m$ be the memory size. The inference-time computational complexity for MemReasonr can be estimated by the encoder complexity $\mathcal{O}(H_1((EL^2 + L_1^2)d_1 + (EL + L_1)d_1^2))$, temporal encoding complexity $\mathcal{O}(Ed^2)$, memory operation complexity $\mathcal{O}(Edm^2)$, decoding complexity $\mathcal{O}(H_2(|P_a|^2 d_2 + |P_a|d_2^2))$, and broadcasting complexity $\mathcal{O}(d_1 dE)$ and $\mathcal{O}(d_2 dH_2)$. For a typical GPT decoding, the inference-time computational complexity is $\mathcal{O}(H_2((EL + L_1)^2 d_2 + (EL + L_1)d_2^2))$.

To provide a more direct comparison, we give in Table 6 the inference cost measured in seconds per input for evaluating with BABILong in comparison to the base decoder (gpt2-large). We note that gpt2-large does not support context lengths longer than 1024 tokens. Overall, we observe that the increase in inference time for MemReasoner is very small for 0k and MemReasoner is more efficient for 1k context length. This is because of utilizing the latent encodings of context, performing one-shot write to the memory, and executing multiple hops over that memory in latent space.

## A.4 COMPARISON TO DECODER-ONLY LMS THAT SUPPORT LONG CONTEXTS

We experiment with Qwen2.5-0.5B and Qwen2.5-1.5B models both of which are decoder-only LMs that support long context windows (up to 128k tokens). The performance of Qwen models on bAbi

| Model type | 0k | 1k | 2k | 4k | 8k | 16k | 32k | 64k | 128k |
|---|---|---|---|---|---|---|---|---|---|
| gpt2-large | 0.28 | 1.13 | - | - | - | - | - | - | - |
| MemReasoner | 0.30 | 0.33 | 0.40 | 0.61 | 0.98 | 1.94 | 3.26 | 11.25 | 13.77 |

Table 6: The inference cost measured in seconds per input on BABILong.

| Model type | Task 1 | Task 2 |
|---|---|---|
| Qwen2.5-0.5B (bAbi) | **100** | 96 |
| Qwen2.5-1.5B (bAbi) | 99.9 | 98.9 |
| MemReasoner-1.4B (bAbi) | **100** | **100** |

Table 7: Performance on bAbi tasks. Best model is highlighted in bold. GPT-3 (=text-davinci-003) baselines are from (Yang et al., 2023). Finetuning data, if any, seen by a model is specified within parentheses.

| Model type | Avg. ≤ 8k | Avg. ≥ 16k | 0k | 1k | 2k | 4k | 8k | 16k | 32k | 64k | 128k |
|---|---|---|---|---|---|---|---|---|---|---|---|
| Qwen2.5-0.5B (bAbi) | 45.4 | - | 94 | 66 | 34 | 23 | 10 | 3 | 1 | - | - |
| Qwen2.5-1.5B (bAbi) | 61.6 | - | **100** | 81 | 57 | 42 | 28 | 32 | 18 | - | - |
| MemReasoner-1.4B (bAbi) | **84.6** | **68.5** | 99 | **91** | **83** | **76** | **74** | **71** | **68** | **70** | **65** |

Table 8: BABILong Task 1 Results - Qwen family models.

| Model type | Avg. ≤ 8k | Avg. ≥ 16k | 0k | 1k | 2k | 4k | 8k | 16k | 32k | 64k | 128k |
|---|---|---|---|---|---|---|---|---|---|---|---|
| Qwen2.5-0.5B (bAbi) | 57.8 | - | 96 | **76** | 59 | 39 | 19 | 11 | 3 | - | - |
| Qwen2.5-1.5B (bAbi) | 46.6 | - | 99 | 67 | 32 | 25 | 10 | 6 | 2 | - | - |
| MemReasoner-1.4B (bAbi) | **60.6** | **18.5** | **100** | 73 | **61** | **46** | **23** | **20** | **19** | **17** | **20** |

Table 9: BABILong Task 2 Results - Qwen family models.

Task 1 and Task 2 is similar to the best in MemReasoner (Table 7).Overall, we find that MemReasoner is able to achieve better length generalization to long contexts compared to Qwen2.5 (Tables 8 and 9). For location changes, we find that MemReasoner outperforms both Qwen2.5-0.5B and Qwen2.5-1.5B for Task 1, but for Task 2 Qwen2.5-1.5B outperforms MemReasoner (Table10). For task generalization (Table 11), Qwen models perform best at shorter BABILong samples, however MemReasoner excels for demanding $\geq 2k$ BABILong lengths.

## A.5 EXTENSION TO GPTJ-6B

MemReasoner is a model-agnostic way to augment current decoder-only LLMs with dynamically updatable memory. Via end-to-end training, the architecture learns to write the latent encodings in a fixed-size memory, order them in their order of appearance in the context, and perform multiple hop over that context and update the latent query accordingly. The decoder learns a differentiated attention mechanism to the readout from the memory, to accurately generate the final answer and supporting facts (intermediate hops). Below, we provide the results when we train a GPTJ-6B decoder with MemReasoner training protocol, suggesting more or less similar performance compared to MemReasoner-1.3B.

## A.6 BEYOND BABI DATASET

In this section, we explore the generalization of MemReasoner on another dataset, variable tracking (VT) from RULER (Hsieh et al., 2024). In the VT task, the model is given context with lines with information about variable value assignment such as "VAR AAAAA = 16438" or "VAR BBBBB = AAAAA" and the model is prompted to obtain all variables with a specific value. Variable names have the format of 5 repeating letters randomly sampled from the alphabet. We train and evaluate with chains of length 2, 4, 6, 8 , and 10 and return the average accuracy over all chain lengths for the 1 hop and 2 hop VT tasks. In order to pad the context for lengths 1k, 4k, and 16k, we follow the approach taken from RULER of padding with the sentence "The grass is green. The sky is blue. The sun is yellow. Here we go. There and back again.\n" until the context reaches the desired length. This noise is not present during training and the 0k data follow the same distribution as the training data.

| Model type | Task 1 | Task 2 |
|---|---|---|
| Qwen2.5-0.5B (bAbi) | 44.2 | 14.5 |
| Qwen2.5-1.5B (bAbi) | 75.2 | **63.5** |
| MemReasoner-1.4B (bAbi) | **87.2** | 52.7 |

Table 10: Robustness to location changes in bAbi test set.

| Model type | 0k | 1k | 2k | 4k |
|---|---|---|---|---|
| Qwen2.5-0.5B (bAbi) | 97 | **71** | 47 | 32 |
| Qwen2.5-1.5B (bAbi) | **100** | 58 | 36 | 18 |
| MemReasoner-1.4B (bAbi) | 83 | 58 | **50** | **45** |

Table 11: Performance on bAbi task 2 → BABILong task 1 generalization.

Since VT asks for all variables with a specific value, for MemReasoner, we take all unordered readouts of the model and pass them individually to the decoder to get the variables from each reasoning hop, and then concatenate these variables in order to obtain the final answer. For RMT, we train with 2 segments, with segment size set to the median length on the train dataset. From Table 13 and Table 14, we observed that it is difficult to train RMT with 2 segments for the 2-hop VT task, RMT can easily learn a shortcut and have high accuracy on the training set, but does not generalize well to the test set at 0k length and performance degrades further at longer context length. Larimar also learned short cuts on 2-hop VT tasks and could not perform well on test sets.

## A.7 COMPARISON WITH TRADITIONAL NON-LLM MEMORY NETWORKS

We have performed additional experiments to evaluate the performance of MemN2N (Sukhbaatar et al., 2015), which we trained on bAbi task 1 and task 2 data with final answer supervision and achieved 100% test accuracy on both. The results are summarized in the following tables and demonstrate the lack of generalization ability of MemN2N compared to MemReasoner.

## A.8 ABLATION STUDIES

### A.8.1 MEMORY

In Table 16, we conduct the ablation study on the episodic memory module in MemReasoner on bAbi and BABILong, task 1 and 2. Specifically, MemReasoner w/o memory module uses the same architecture of encoder and decoder (BERT-Large and GPT2-Large respectively) but does not use the memory module for encoding the context. Instead, the MemReasoner w/o memory uses the encoder to encode only the question and this is passed in to the decoder as kv-cache. Additionally, the context and question are passed to the decoder as part of the prompt with the format:

```
Context:
{context}
Question:
{question}
Answer:
```

where {context} and {question} represent the context and the question for the datapoint. We train the model with reconstruction loss to ensure that the model is able to fill in the answer given this prompt and with autoencoding loss on the pretraining dataset (see last term of Equation 1) in order to reduce overfitting on bAbi data. We train MemReasoner w/o memory module for 5 epochs.

MemReasoner w/o memory module trained on bAbi task 1 obtains almost perfect accuracy on bAbi task 1 and BABILong task 1 0k. However, its generalization ability to long context (BABILong 1k and 2k) is much inferior to MemReasoner (MemReasoner\memory 0% vs. MemReasoner 91% on BABILong 1k). Similar trends can also be seen from bAbi task 2 trained MemReasoner\memory, implying the significance of the episodic memory module and the operations around it in MemReasoner.

### A.8.2 TEMPORAL ENCODING

In Table 17, we experiment with different temporal encoding schemes, including non-parametric method (Positional Encoding) and parametric method (GRU). In the table, we show MemRea-

| Model type | 0k | 1k | 2k | 4k | 8k | 16k | 32k | 64k | 128k |
|---|---|---|---|---|---|---|---|---|---|
| Task 1 | 98 | 82 | 77 | 65 | 60 | 68 | 70 | 65 | 67 |
| Task 2 | 98 | 65 | 50 | 34 | 35 | 32 | 22 | 27 | 30 |

Table 12: Performance of MemReasoner with a GPTJ-6B decoder on BABILong.

| Model type | 0k | 1k | 4k | 16k |
|---|---|---|---|---|
| RMT-.77B (VT) | 100 | 5.7 | 5.0 | 4.5 |
| Larimar-1.3B (VT) | 92.5 | 92.5 | 94.0 | 93.6 |
| MemReasoner-1.4B (VT) | 99.9 | 100.0 | 99.9 | 99.9 |

Table 13: Single hop variable tracking results.

soner's accuracy on BABILong Task 1. It can be seen that GRU encoding has significant advantage over Positional Encoding, with much slower decay in the accuracy as the context length increases. Additionally, though showing higher accuracy compared with Positional Encoding, uni-directional GRU's accuracy decreases faster than bi-directional GRUs. Since 1-layer bi-directional GRU has similar performance with 2-layer bi-directional GRU, we choose the lighter model and use 1-layer bi-directional GRU throughout the experiments in this paper.

### A.8.3 QUERY UPDATE $\alpha$

In Table 18, we exploit test-time inference hyper-parameter $\alpha$ and its effect in reasoning tasks' performance. We draw inspiration from (Kollias et al., 2024), where authors investigated the effect of scaling readout vectors to improve generation quality. In Line 20 of Algorithm 1, when using an $\alpha > 1$, we equivalently scale up the readout vectors which greatly help our generalization to Task 1 BABILong according to Table 18 (e.g. from 14% to 45% on 4k context token task).

### A.9 EFFECT OF ARBITRARY NUMBER OF HOPS WITH WEAKER SUPERVISION

Table 19 shows performance of MemReasoner that is trained with weaker supervision on bAbi task 2, and is tested on BABILong task 2 test set. In this case, during training an arbitrary number (5) of hops was used, together with supervision only on the final supporting fact and the final answer. While performance on longer samples drops compared to the model trained with full supervision, the model generalizes well on 1k tokens long BABILong samples compared to other baselines (see Table 3. This direction will be further explored in future work.

### A.9.1 TRAINING EPOCHS

In Table 20, we evaluate MemReasoner's performance when fine-tuned on bAbi task 2 as a function of the number of training epochs. Specifically, with fewer epochs, MemReasoner demonstrates stronger robustness to location change, reaching an accuracy of 79% at the 66th epoch, which decreases to around 50% as the training continues (at 100/200th epoch). On the other side, MemReasoner's accuracy on shorter context tasks in BABILong Task 1 and 2 (i.e. 0-4k) improves as the training continues.

### A.10 LIMITATIONS AND FUTURE WORK

The current work is limited to testing the MemReasoner framework on synthetic reasoning tasks only. Future work will extend the framework to evaluating reasoning generalization on natural language datasets. Another potential direction is extending MemReasoner to scenarios with weaker and noisy supervision.

| Model type | 0k | 1k | 4k | 16k |
|---|---|---|---|---|
| RMT-.77B (VT) | 74.6 | 5.7 | 1.3 | 0.2 |
| Larimar-1.3B (VT) | 0.1 | 0 | 0.1 | 0 |
| MemReasoner-1.4B (VT) | 98.4 | 97.6 | 97.0 | 98.0 |

Table 14: Two hop variable tracking results.

| Model type | 0k | 1k | 2k |
|---|---|---|---|
| BABILong Task 1 | 100 | 36 | 15 |
| BABILong Task 2 | 100 | 54 | 21 |
| bAbi task 2 → BABILong task 1 | 53 | 26 | 19 |

Table 15: Performance of MemN2N.

| Model type | Task 1 | 0k | 1k | 2k | Task 2 | 0k | 1k | 2k |
|---|---|---|---|---|---|---|---|---|
| MemReasoner\memory | **100** | **100** | 0 | - | 99.3 | **100** | 29 | - |
| MemReasoner | **100** | 99 | **91** | **83** | **100** | **100** | **73** | **61** |

Table 16: Ablation study on the episodic memory

| Encoding scheme | 0k | 1k | 2k |
|---|---|---|---|
| Positional Encoding | **100** | 27 | 20 |
| 2-layer bi-directional GRU | **100** | 90 | 80 |
| 2-layer uni-directional GRU | 94 | 75 | 61 |
| 1-layer bi-directional GRU | 99 | **91** | **83** |

Table 17: Ablation study on the temporal encoding schemes.

| Query update $\alpha$ | Task 2 bAbi location change | Task 2 BABILong | | | | | | | | | Task 1 BABILong | | | |
|---|---|---|---|---|---|---|---|---|---|---|---|---|---|---|
| | | 0k | 1k | 2k | 4k | 8k | 16k | 32k | 64k | 128k | 0k | 1k | 2k | 4k |
| 1 | 52.6 | **100** | 46 | 25 | 18 | 18 | 13 | 16 | 12 | 13 | 78 | 21 | 17 | 14 |
| 4 | **54.2** | **100** | **73** | **61** | **46** | 26 | 22 | **19** | **19** | **27** | **83** | 47 | 44 | 40 |
| 8 | 52.7 | **100** | **73** | **61** | **46** | 23 | 20 | **19** | 17 | 20 | **83** | **58** | **50** | **45** |

Table 18: Ablation study on the query update parameter $\alpha$.

| Model type | 0k | 1k | 2k | 4k |
|---|---|---|---|---|
| RMT-.14B (bAbi) | 97 | 31 | 19 | 16 |
| RMT-.77B (bAbi) | 100 | 36 | 21 | 27 |
| Mamba-.13B (bAbi) | 64 | 10 | 3 | 3 |
| Mamba-1.4B (bAbi) | 94 | 44 | 15 | 5 |
| Larimar-1.3B (bAbi) | 42 | 41 | 29 | 22 |
| MemReasoner (full) | 100 | 73 | 61 | 46 |
| MemReasoner (weak) | 100 | 58 | 31 | 22 |

Table 19: Comparison of MemReasoner trained with full supervision with MemReasoner(weak) on BABILong task 2 samples, where the weak supervision considers an arbitrary five hops and only supervision on final supporting fact and final answer.

| #epochs | Task 2 bAbi location change | Task 2 BABILong | | | | | | | | | Task 1 BABILong | | | |
|---|---|---|---|---|---|---|---|---|---|---|---|---|---|---|
| | | 0k | 1k | 2k | 4k | 8k | 16k | 32k | 64k | 128k | 0k | 1k | 2k | 4k |
| 66 | **78.0** | 99 | 70 | 54 | 30 | 27 | 23 | 17 | **18** | 17 | 58 | 51 | 45 | 37 |
| 100 | 47.3 | **100** | 70 | 57 | 38 | **28** | **31** | **25** | 12 | 19 | 82 | **58** | **50** | **46** |
| 200 | 52.7 | **100** | **73** | **61** | **46** | 23 | 20 | 19 | 17 | **20** | **83** | **58** | **50** | 45 |

Table 20: Ablation study on the number of training epochs

