# OpenReview forum: "MEMREASONER: A MEMORY-AUGMENTED LANGUAGE MODEL ARCHITECTURE FOR MULTI-HOP REASONING"
_ICLR.cc/2025/Conference — Submitted to ICLR 2025_

### Official Review · Reviewer_PgX8 · 2024-11-02

**Soundness:** 2
**Presentation:** 1
**Contribution:** 3
**Rating:** 3
**Confidence:** 5

**Summary:**

This paper proposes MemReasoner, a method that augments language models with memory and is designed to perform reasoning over long contexts. The authors aim to enable models trained on short sequences to generalize and work effectively on longer sequences without retraining. To achieve this, they build on the Larimar model, which augments language models with episodic memory, and enhance it with two key modifications: 1. An iterative mechanism to read information from memory and update the query accordingly. 2. Adding explicit information about temporal ordering of facts within the context (via position embeddings or BiGRU). The approach is evaluated on the bAbI dataset and its long sequence variant BABILong, demonstrating the ability to perform on longer sequences while being trained on short ones, and improving performance over existing methods in this setup.

**Strengths:**

- General idea of iterative reading from memory for multi-hop tasks is clear and reasonable.
- The motivation to address the challenge of generalizing from short to long sequences without extensive retraining is strong and relevant, and addresses a key limitation of current methods.
- Results show that the proposed approach is indeed able to generalize to longer sequences being trained on short only.
- The evaluation setup, using the bAbI dataset and its longer version, is clear and suitable for testing the ability of the method to solve inference tasks over varying sequence lengths.

**Weaknesses:**

- Giving that idea is clear, unfortunately Section 3 that describes the proposed method is hard to read and lacks clarity. I believe clarity of presentation of the proposed method should be largely improved. Details are provided in the suggestions section.
- Lack of comparison with Transformer models. The authors do not evaluate the performance of transformer-based LMs that support long contexts (e.g., LLama-3.2, Phi-3.5, or Qwen-2.5). These models could be fine-tuned on bAbI and evaluated on BABILong, providing relevant baselines for comparison. Including such evaluations would contextualize the performance of MemReasoner relative to current state-of-the-art models.
- Generalization of the proposed method to tasks other than bAbI is not supported. There are other datasets that require multi-hop reasoning, such as MultiHopQA, MuSiQue, HotPotQA. Their context could be extended by extracting relative paragraphs from e.g. Wikipedia.
- The paper does not provide empirical evaluations of the method’s inference time or memory consumption compared to other methods. However, authors provide theoretical time complexity in Appendix A.2.

**Questions:**

Suggestions on improving presentation:
- Figure 2 is hard to comprehend – it is not clear where to start from. I would suggest making two schemes: 1. conceptual scheme of the method. 2. a detailed one with how the memory module and the query are iteratively updated.
- Similar problem with Section 3 - overloaded preliminary section hindering comprehension. Section 3.1 contains too many details on how memory read/write operations are constructed, making it challenging to grasp the overall picture. Splitting the explanation into two parts – a conceptual overview and detailed descriptions of the memory module and reasoning within it – would help readers understand the main ideas before diving into technical specifics. As the method is based on the Larimar model, a dedicated section describing Larimar would greatly improve clarity.
- L292-296 present formulas for the loss components that are difficult to match with their explanations. Defining each loss component separately and explaining their roles can improve clarity and comprehension.
- How do P_a and P_s look like in L270–271? P_a is later defined in L371–372. Providing these definitions earlier would help improve clarity.
- Typos and Minor Issues
  - L315: "facts from bAbI is distributed" -> are
  - L318–319: missing space "tokens.For"

Comments:
- One of the results is that RMT being trained on short sequences (single segment) can not generalize to longer ones (multiple segments). RMT inherently can not generalize from 1 segment to multiple segments. To learn to use memory (to pass information sequentially) it needs at least 2 segments. It was shown in RMT paper and generalization to larger lengths is possible with curriculum learning procedure (“Beyond Attention: Breaking the Limits of Transformer Context Length with Recurrent Memory”). So this makes comparison with RMT not so fair.

Questions:
- Table 1 and 2 miss results for RMT on 64k and 128k. Authors mention that it “means unavailable due to out of memory errors or maximal input length constraints” (L426). However, RMT has constant memory consumption as it is required to pass only memory states between segments, and there is no need to all keep intermediate outputs. So, it is not clear why these values are missing? However, I acknowledge that they would be low.
- Is it possible to train MemReasoner on longer sequences and compare it with RMT/Mamba which has also been trained on longer sequences? Will the performance of MemReasoner continue to improve?
- L182 - what does “mimicking Bayesian inference” mean here? How is it motivated and why is it mimicking? It is unclear how this concept is applied and whether it is key to constructing an effective memory module.
- After the last readout operation, the z_i most similar to z_r is selected. Z_i, representing a single fact from bAbI, goes to a decoder. It seems to be sufficient for QA1-2 tasks to get the correct answer from bAbI, but would it be sufficient for general tasks to extract only one fact from memory? Why not to use z_r itself?
- The paper does not clearly explain whether the number of memory read operations correlates with the number of fact hops required in reasoning tasks. For example, in QA2 tasks, a fixed number of two read operations is used. Is this number optimal? Would using only one read operation be insufficient to retrieve the answer? Providing an analysis or justification for the chosen number of read operations would clarify this aspect.

Personally, I like the idea and direction of the work and find it interesting. However, I think it could be improved a lot.

---

> ### Author Response · Authors · 2024-11-26
> **Official Comment by Authors (1/3)**
>
> We thank the reviewer for their constructive feedback and for recognizing our idea and the direction of the work as "interesting", "clear and reasonable", and "strong and relevant".  Please see our detailed responses below:
>
> > Lack of comparison with Transformer models
>
> Please refer to global response section on "Comparison to decoder-only LMs that support long contexts". We have now aded Qwen2.5 baslines as requested by the reviewer, which further establishes that current LLMs with long-context training does not enable robust reasoning generalization.
>
>  > Generalization of the proposed method to tasks other than bAbI is not supported. There are other datasets that require multi-hop reasoning, such as MultiHopQA, MuSiQue, HotPotQA. Their context could be extended by extracting relative paragraphs from e.g. Wikipedia.
>
> We thank the reviewer for this specific comment. We would like to clarify that although traditional multihop QA like HotpotQA, MusiQue, and 2WikiMultihopQA can be applied to evaluate the multi-step reasoning ability, existing studies suggest that those datasets often fail short while evaluating LM's logical reasoning abilities. One underlying reason is the interference from the parametric knowledge, likely caused by the training data contamination during LLM pretraining phase, as all of these benchmarks include data sourced from wikipedia. As a result, LLM may not use the non-parametric knowledge provided in the context while answering (as shown in Wu, et al. https://arxiv.org/abs/2402.11924v4. Another important factor to be considered is the presence of reasoning shortcuts in the task samples themselves, which does not ensure language models are actually being required to perform multi-hop processing over the context, as shown for HotpotQA in Jiang, et al. 2019 https://arxiv.org/abs/1906.07132. Since the goal of our study is to propose and evaluate language model architectures that do indeed perform more than one hop to solve a logical reasoning task in the absence of real-world knowledge, we avoid the potential data contamination issue affecting model's reasoning ability evaluation by training and testing on synthetic logical reasoning tasks. Further, we subject the model to an even more difficult scenario, where the core reasoning facts are diffused with distracting task-irrelevant natural text, the datasets suggested by the reviewer do not fit that framework. And to avoid the issue of model taking reasoning shortcuts instead of performing multiple hops, we focus on classical logical reasoning tasks such as babi and show evaluation on test distributions that are significantly different from the training samples.
>
> However, we do provide additional experimental results on variable tracking tasks (both single-hop and two-hop) from the RULER benchmark, since we agree that it is important to show the generality of MemReasoner, when applied to other long-context tasks.  Please refer to the global response section on "Additional datasets" for these results.
>
> > The paper does not provide empirical evaluations of the method’s inference time or memory consumption compared to other methods.
>
> Please refer to the global response section on "Computational cost of MemReasoner".
>
> > Suggestions on improving presentation
>
> We truly appreciate the reviewer's feedback on improving presentation of the MemReasoner framework by modifying Figure 2 and section 3.1. As suggested, we are revising the section 3 and the Figure 2 in manuscript by beginning with a conceptual description and then providing further details of the model and method. We are also adding a dedicated section on Larimar and explicitly elaborated on each loss term.

---

> ### Author Response · Authors · 2024-11-26
> **Official Comment by Authors (2/3)**
>
> > RMT training with at least 2 segments
>
> We thank the reviewer for pointing out the importance of training with multiple segments using a curriculum learning procedure. In order to train with more segments while exposing the model to only bAbi data, we reduce the segment size to 64 for task 1 and 128 for task 2.  This leads to 2 segments in training for task 1 and 2-4 segments in training for task 2.  In order to mimic the curriculum learning process described in the referenced work, we filter the data so that we train with inputs with token length up to the segment size for 10 epochs, up to 2 times the segment size for another 10 epochs, and so on.  We present updated values for RMT below, which will be added to the revised manuscript:
>
>
> BABILong Task 1:
> | Model | 0k | 1k | 2k | 4k | 8k | 16k | 32k | 64k | 128k |
> | -------- | -------- | -------- | -------- | -------- | -------- | -------- | -------- | -------- | -------- |
> | RMT-.14B(bAbi) |   96   |   4   |  26  |   19   |   19   |  12  |   22  |  12    | 16   |
> | RMT-.77B(bAbi) |   99   |   27   |  21  |   25   |  14    |  14  |   19   |   16   |  18 |
>
> BABILong Task 2:
> | Model | 0k | 1k | 2k | 4k | 8k | 16k | 32k | 64k | 128k |
> | -------- | -------- | -------- | -------- | -------- | -------- | -------- | -------- | -------- | -------- |
> | RMT-.14B(bAbi) |   97   |   31   |  19  |   16   |   20   |  12 |  12    |  14    |  10 |
> | RMT-.77B(bAbi) |   100   |   36   |  21  |   27   |  22    |  18  |   23   |   13   | 16  |
>
>
> Results for RMT-.77B on babi + location swap
>
>
> | Model | Task 1 | Task 1 (loc swap) | Task 2 |Task 2 (loc swap) |
> | -------- | -------- | -------- |-------- |-------- |
> | RMT-.77B  |   97.7   |   44.7   |   97.5  |  0.6 |
>
> RMT-.77 Task 2 -> Task 1 generalization
>
> | Model | 0k | 1k | 2k | 4k |
> | -------- | -------- | -------- |-------- | -------- |
> | RMT-.77B     |   100   |   19   | 20 | 12 |
>
> > Table 1 and 2 miss results for RMT on 64k and 128k.
>
> Thank you for pointing this out, we realized that in our evaluation we were storing intermediate states which was causing the out of memory error.  Please refer to the tables above with training RMT with multiple segments as these tables have values for all BABILong lengths.
>
> > Is it possible to train MemReasoner on longer sequences and compare it with RMT/Mamba which has also been trained on longer sequences? Will the performance of MemReasoner continue to improve?
>
> Please refer to the global response section on "Finetuning on long data".
>
> > 182 - what does “mimicking Bayesian inference” mean here? How is it motivated and why is it mimicking? It is unclear how this concept is applied and whether it is key to constructing an effective memory module.
>
>
> We thank the reviewer for raising this point. We plan to add the following discussion in the paper to clarify this point as follows: MemReasoner follows the earlier works on Kanerva Machine (Wu et al., 2018a), which is inspired by Kanerva’s sparse distributed memory model (Kanerva, 1988), where the memory is viewed as a global latent variable in a generative model. In this framework, the goal is to learn a memory dependent data prior and learnable addresses, where the memory update and read/write are considered as Bayesian inference, i.e., the posterior parameters are updated as new data arrives. Later, Pham et al., 2021, reformulated the encoding of new memories and decoding data from memories from Bayesian updates to an equivalent minimization problem, which essentially amounts to solving a linear system of equations,  efficiently done via computing matrix pseudo inverses. MemReasoner follows this approach of reformulation of the Bayesian update as finding the least-squares solution to a linear system for updating the memory parameters  (see Algorithm 1).
>
> Yan Wu, Greg Wayne, Alex Graves, and Timothy Lillicrap. "The Kanerva Machine: A Generative Distributed Memory." In International Conference on Learning Representations. 2018.
>
> Pentti Kanerva. Sparse distributed memory. MIT press, 1988.
>
> Kha Pham, Hung Le, Man Ngo, Truyen Tran, Bao Ho, and Svetha Venkatesh. "Generative pseudo-inverse memory." In International Conference on Learning Representations. 2022.

---

> ### Author Response · Authors · 2024-11-26
> **Official Comment by Authors (3/3)**
>
> >The paper does not clearly explain whether the number of memory read operations correlates with the number of fact hops required in reasoning tasks. For example, in QA2 tasks, a fixed number of two read operations is used. Is this number optimal? Would using only one read operation be insufficient to retrieve the answer? Providing an analysis or justification for the chosen number of read operations would clarify this aspect.
>
> In our experiments, the number of memory read operations corresponds to the number of hops. The reason for this is that in the MemReasoner training pipeline, we enforce that each read operation retrieves the relevant supporting fact for that hop via the reconstruction of supporting facts and ordering losses. To illustrate, consider the context ["Mary is in the kitchen.", "Mary got the apple there."] and the query "Where is the apple?".  In training, we ensure that the first read retrieves the sentence "Mary got the apple there.", the readout is then used to update the query so that the query now has some encoded information about "Mary", and then we ensure that a second read with this updated query retrieves "Mary is in the kitchen.", which is then given to the decoder to generate the answer.
>
> >After the last readout operation, the z_i most similar to z_r is selected. Z_i, representing a single fact from bAbI, goes to a decoder. It seems to be sufficient for QA1-2 tasks to get the correct answer from bAbI, but would it be sufficient for general tasks to extract only one fact from memory? Why not to use z_r itself?
>
> First, to clarify, the readout $z_r$ also corresponds to a single fact in bAbI that was written to memory. The difference between $z_r$ and $z_i$ is that $z_i$ is the encoding prior to passing the lines through the GRU, so $z_r$ contains information about the relative ordering of the line in the context while $z_i$ does not. This relative order learning is important, as in the bAbi tasks the agents's location can be updated within one sample (i.e., one episode) See Figure 1, Right as an example, where Sandra's location is updated from bedroom to hallway. Therefore, it is crucial that the decoder's response is built on top of the most relevant (aka, recent) location of Sandra. Since the decoder only needs information about what is present in the most recent retrieved bAbi fact and not the relative order of it, we chose to pass $z_i$ to the decoder in the MemReasoner design.  Passing $z_r$ to the decoder can cause the decoder to overfit to the ordering information in $z_r$ which hinders length generalization.
>
> As noted by the reviewer, some reasoning tasks may require outputting information from multiple facts as opposed to just the final supporting fact.  For example, for the variable tracking task (see global response "Additional Datasets"), the prompt asks for all variables with a certain value (with the number of variables in the answer corresponding to the number of hops) and each fact specifies only the value of a single variable.  In order to handle this case, we can take all readouts $z_{r_1}...z_{r_h}$ where $h$ is the number of hops, map each readout to unordered encodings $z_{i_1}...z_{i_h}$ and then these are individually passed to the decoder and the outputs from the decoder are then aggregated as the final list of $h$ variables.

---

> > ### Comment · Reviewer_PgX8 · 2024-11-27
> >
> > I thank the authors for their detailed responses and the clarifications provided. The responses partially address some of the raised concerns. However, there are a few key points remain:
> >
> > On generalization:
> >
> > MemReasoner relies on supervision for each hop and predefined number of hops, which limits its generalizability.
> >
> > While fixing the number of hops is a practical solution for specific tasks, the need for explicit supervision at each hop raises concerns about its applicability to diverse tasks where such supervision is unavailable or the number of hops varies. In real-world multi-hop reasoning scenarios, the number of required hops may be task-dependent and not always clearly defined.
> >
> > In contrast, the baseline models do not rely on knowledge of the number of hops or intermediate supervision at each reasoning step. It would be highly valuable to explore how MemReasoner performs under relaxed supervision - such as using only the final task labels without explicit hop-level guidance - and whether it can handle tasks requiring varying numbers of hops.
> >
> > While the authors argue that current benchmarks like HotpotQA, MusiQue, and 2WikiMultihopQA suffer from shortcuts and data contamination, approaches that explicitly support multi-hop reasoning may still benefit more from evaluation on these datasets than methods without such explicit support, using same base models. Demonstrating performance on real-world benchmarks, even if imperfect, could strengthen the case for the practical utility and generalizability of MemReasoner.
> >
> > On finetuning on long data:
> >
> > Training on bAbI and evaluating on BABILong may present challenges for certain baselines, such as Mamba and RMT, due to the significant train-test mismatch in sequence lengths. There is no need to train on extremely long sequences, such as 80k tokens. Instead, training on more reasonable sequence lengths, such as 4k-8k, could provide a practically feasible approach that ensures a fairer comparison while still demonstrating the benefits of the memory reasoning module. For MemReasoner, which relies on additional supervision, this mismatch may favor its performance due to its dependence on task-specific design choices, thereby limiting the scope of fair comparisons between models.

---

> > > ### Author Response · Authors · 2024-12-02
> > > **Further clarification by Authors (1/2)**
> > >
> > > We would like to thank the reviewer for responding to our rebuttal! We hope our further clarification and additional results below can make our points clearer.
> > >
> > > > On generalization:
> > > MemReasoner relies on supervision for each hop and predefined number of hops...... and whether it can handle tasks requiring varying numbers of hops.
> > >
> > > We thank the reviewer for considering our response and clarifications provided. Regarding the point on generalization, indeed, we want to make this point in our paper, that **internal supervision on intermediate processing steps, together with the supervision on the final reasoning outcome, that explicitly utilizes the latent memory operations, is a means to improve reasoning generalization abilities of LLMs**, and MemReasoner is a demonstration of that. The current memory-based baselines show that supervision on final outcome alone is not sufficient for the task and the models may learn a reasoning shortcut. To address the point brought up by the reviewer that the baselines do not have knowledge of intermediate steps/hops, we have now further added a new experiment where we encouraged the RMT baseline (finetuned on bAbi) to additionally retrieve the correct supporting fact by completing "Context:{context}\nQuestion:{question}\nSupporting facts:"". We denote this setting as RMT w/ SP (supporting fact supervision), which obtains 100% and >97% accuracy on bAbi training and test sets, respectively. However, the improvement by adding this simple supervision using supporting facts with RMT is not obvious on generalization testbeds (see the results below). As it only guarantees that the model recognizes the true supporting fact(s), but the decoding is still not explictly conditioned on the retrieved supporting fact (or its latents). As a result, RMT w/ SP does not generalize to longer BABIlong samples within the same task or across task (task 2-->task 1) or when answer locations are different in the test data from the training samples for a given task.  These experiments show that simple access to intermediate reasoning steps is not sufficient for achieving reasoning generalization. Therefore, future work shall consider combining algorithmic (explicit hops) with architectural (memory module) implicit biases and extend that to recurrent models like RMT to fully unleash the potentials with supporting fact supervision.
> > >
> > > BABILong Task 1:
> > > | Model |  $\le$ 8k| $\ge$ 16k| 0k | 1k | 2k | 4k | 8k | 16k | 32k | 64k | 128k |
> > > | -------- | -------- | -------- | -------- | -------- | -------- | -------- | -------- | -------- | -------- | -------- | -------- |
> > > | RMT-.77B(bAbi) |  37.2 | 16.7 | 99   |   27   |  21  |   25   |  14    |  14  |   19   |   16   |  18 |
> > > | RMT-.77B(bAbi) w/ SP | 36.8 | 22 | 100 | 23 | 31 | 11 | 19 | 24 | 21 | 22 | 21
> > >
> > > BABILong Task 2:
> > > | Model | $\le$ 8k| $\ge$ 16k| 0k | 1k | 2k | 4k | 8k | 16k | 32k | 64k | 128k |
> > > | -------- | -------- | -------- | -------- | -------- | -------- | -------- | -------- | -------- | -------- | -------- | -------- |
> > > | RMT-.77B(bAbi) |  41.2  | 17.5 |   100   |   36   |  21  |   27   |  22    |  18  |   23   |   13   | 16  |
> > > | RMT-.77B(bAbi) w/ SP |  40.6  | 20.3 | 100 | 41 | 22 | 21 | 19 | 22 | 21 | 18 | 20
> > >
> > > Results for RMT-.77B on babi + location swap
> > >
> > > | Model | Task 1 | Task 1 (loc swap) | Task 2 |Task 2 (loc swap) |
> > > | -------- | -------- | -------- |-------- |-------- |
> > > | RMT-.77B(bAbi)  |   97.7   |   44.7   |   97.5  |  0.6 |
> > > | RMT-.77B(bAbi) w/ SP | 100 | 31.3 | 96.8 | 0
> > >
> > > RMT-.77 Task 2 -> Task 1 generalization
> > >
> > > | Model | 0k | 1k | 2k | 4k |
> > > | -------- | -------- | -------- |-------- | -------- |
> > > | RMT-.77B(bAbi)     |   100   |   19   | 20 | 12 |
> > > | RMT-.77B(bAbi) w/ SP | 95 | 21 | 22 | 16
> > >
> > > We have shown in the paper the MemReasoner generalizes from task 2 (2-hop) to task 1 (1-hop), where the test samples correspond to BABIlong distribution, or within the same task where the test samples contain different answer locations. We have also added new experiments to show that MemReasoner can also operate under relaxed supervision conditions (1) when only partial supporting facts are available and further (2) when the number of hops is not explicitly known, and still shows length generalization under those conditions. Future work will further explore this angle, where the supervision on intermediate steps and knowledge of number of hops for the task will be varied.
> > >
> > > BABILong Task 2:
> > > | Condition | $\le$ 8k| $\ge$ 16k| 0k | 1k | 2k | 4k | 8k | 16k | 32k | 64k | 128k |
> > > | -------- | -------- | -------- | -------- | -------- | -------- | -------- | -------- | -------- | -------- | -------- | -------- |
> > > | (1) partial (second only) supporting fact | 46.4 | 13.5 | 99 | 56 | 38  | 23 | 16 | 13 | 15 | 12 | 14
> > > | (1+2) partial (second only) supporting fact + unknown \#hops (set to 5)| 46.2 | 12.8 | 100 | 58 | 31 | 22 | 20 | 17 | 7 | 9 | 18

---

> > > ### Author Response · Authors · 2024-12-02
> > > **Further clarification by Authors (2/2)**
> > >
> > > > While the authors argue that current benchmarks...... could strengthen the case for the practical utility and generalizability of MemReasoner.
> > >
> > > We agree with the reviewer that MemReasoner should be extended to more benchmarks including the real-world ones. We have added new experiments on another multi-hop task, namely variable tracking from another long-context benchmark RULER (Tables 13 and 14 in the updated pdf). Future works will further extend MemReasoner to other benchmarks.
> > >
> > > > On finetuning on long data:
> > > Training on bAbI and evaluating on BABILong...... thereby limiting the scope of fair comparisons between models.
> > >
> > > This is an excellent point. However, the goal of this work is to ensure that MemReasoner does not see any training samples that is a mixture of bAbi and PG-19 corpus. And when trained on bAbi samples alone, together with the additional supervision on intermediate reasoning steps, we wanted to test how well MemReasoner can perform, when compared to RMT and Mamba that are trained on BABIlong distribution and achieve near accurate performance on the exact same test distribution. That is why we test MemReasoner in the setting, where the model (and the baselines) is not exposed to samples from BABIlong distribution during training. We have further clarified this point in the updated pdf, that while training on longer samples from the final distribution (e.g., BABILong) with a supervision on final answer is one way to achieve reasoning generalization over varying input length, a **complementary** approach is to train a model with latent memory operations and the strong supervision on shorter samples using both final answer and intermediate step(s). Of course, both approaches can be combined, which will be considered in future work.

---

### Official Review · Reviewer_N6d6 · 2024-11-03

**Soundness:** 3
**Presentation:** 3
**Contribution:** 3
**Rating:** 6
**Confidence:** 3

**Summary:**

The paper introduces MemReasoner, a memory-augmented LLM designed to improve temporal and multi-hop reasoning over extremely long contexts. MemReasoner integrates an episodic memory module and introduces a novel mechanism for explicitly learning temporal relationships between events and for iteratively reading and updating queries, aiming to address the limitations of existing LLMs in processing and reasoning across long documents. The proposed model is evaluated on the BABILong benchmark, demonstrating superior generalization compared to existing LLMs.

**Strengths:**

- The authors present an architecture that uses a memory module for capturing temporal relationships and enabling multi-hop reasoning, which adds a valuable improvement over standard LLMs.
- MemReasoner demonstrates a robust ability to reason across contexts of up to 128k tokens, outperforming existing LLMs and memory-augmented architectures on the challenging BABILong benchmark.

**Weaknesses:**

- The experiments are largely based on synthetic datasets like BABILong, which, while controlled, may not fully reflect the complexities of real-world language tasks. Extending evaluations to diverse, natural datasets would strengthen the validity of the model’s utility.
- The architecture involves multiple components like GRU-based temporal encoding and iterative query updates, which might increase computational complexity, potentially making it less scalable for broader applications.

**Questions:**

- What improvements does MemReasoner have compared to Larimar?
- Has the model been tested on other multi-hop reasoning datasets or more general NLP tasks?
- How does MemReasoner compare to traditional, non-LLM memory networks? Are there any experiments comparing it to these earlier methods, and what would the outcomes likely be?

---

> ### Author Response · Authors · 2024-11-26
> **Official Comment by Authors (1/2)**
>
> We thank the reviewer for their constructive feedback and for recognizing our work as " a valuable improvement over standard LLMs".  Please see our responses below:
>
> > Extending evaluations to diverse, natural datasets
>
> We thank the reviewer for this specific comment. We would like to clarify why we evaluate MemReasoner on the BABIlong benchmark. First of all, the original bAbi tasks 1 and 2, though simple in nature, provide a good testbed for evaluating LM's logical reasoning abilities, which is even found to be challenging for more powerful GPT-3 model, even when combined with advanced prompting methods like CoTs. The tasks do comprise many basic elements of reasoning, such as order learning among facts and multi-hop reasoning. And, it should be noted that to date, most logical reasoning datasets are synthetic in nature.
>
> Second, BABIlong provides a controlled testbed for testing reasoning generalization over long-context, where core reasoning facts (that are synthetic) are diluted with irrelevant naturally occurring text throughout the context of varying length. This benchmark is found to be challenging to current LLMs and RAG frameworks. With this construct, due to its synthetic nature and careful construction, the interference from LM's parametric knowledge can be kept at a minimum. This is not the case for many real-world natural language reasoning tasks, which are at risk of training data contamination during LLM pretraining phase. As a result, LLM may not use the non-parametric knowledge provided in the context while answering (as shown in Wu, et al. https://arxiv.org/abs/2402.11924v4.). Thus, BABIlong allows us to train the model on classical logical reasoning tasks such as babi and show evaluation on test distributions that are significantly different from the training samples, which is the main goal of this work.
>
> A third reason to be considered is the presence of reasoning shortcuts in the natural language task samples itself, which does not ensure language models actually being required to perform multi-hop processing over the context, as shown for HotpotQA in Jiang, et al. 2019 https://arxiv.org/abs/1906.07132. This is not the case for BABIlong.
>
> To show the generality of MemReasoner across different tasks, we have now added experiments on another task, namely variable tracking, from the RULER benchmark (see the global response section "additional datasets".). The new results on the variable tracking experiments also show MemReasoner outperforming other baselines such as RMT and Mamba.
>
>
> > Computational complexity
>
> Please refer to global response section on "Computational cost of MemReasoner".
>
> > What improvements does MemReasoner have compared to Larimar?
>
> We note that the Larimar architecture was designed for fact-editing in a single-hop setting.  While MemReasoner changes the Larmar architecture and training procedure toward multi-hop reasoning, where learning the relative order of facts within an episode is crucial. In terms of architectural changes, MemReasoner introduces an iterative read with a query update procedure and a temporal encoding module into the Larimar architecture. The temporal encoding module allows us to capture the relative ordering of sentences within an episode, which allows us to perform reasoning on tasks where the ordering of facts within the context matters (for example, bAbi dataset). Additionally, the iterative read and the query update allow the model to perform multi-hop reasoning, with each read operation retrieving the supporting fact for each hop. In terms of training modifications, we introduce additional loss terms into MemReasoner (reconstruction of supporting facts and ordering loss) to ensure that the read operations retrieve the correct supporting fact.

---

> ### Author Response · Authors · 2024-11-26
> **Official Comment by Authors (2/2)**
>
> > How does MemReasoner compare to traditional, non-LLM memory networks? Are there any experiments comparing it to these earlier methods, and what would the outcomes likely be?
>
> We have performed additional experiments to evaluate the performance of MemN2N[1]. The results are summarized in the following tables and demonstrate the lack of generalization ability of MemN2N compared to MemReasoner.
>
>  Table 1 (Performance on bAbi Tasks):
> | Model | Task 1 | Task 2|
> | -------- | -------- | --------|
> | MemN2N (bAbi)|    100   |    100   |
> | MemReasoner-1.4B (bAbi) |    100   |    100   |
>
> Table 2 (BABILong Task 1 Results):
> | Model | 0k | 1k | 2k |
> | -------- | --------     | -------- | -------- |
> |  MemN2N (bAbi) |   100   |   36   |  15  |
> | MemReasoner-1.4B (bAbi) |   99   |   91   |   83 |
>
> Table 3  (BABILong Task 2 Results):
> | Model | 0k | 1k | 2k |
> | -------- | -------- | -------- | -------- |
> | MemN2N (bAbi) |  100    |   54   |  21  |
> | MemReasoner-1.4B (bAbi)              |   100 |   73 |   61 |
>
> Table 4 (Performance on bAbi task 2 → BABILong task 1 generalization):
> | Model | 0k | 1k | 2k |
> | -------- | -------- |-------- |-------- |
> | Mem2N (bAbi) |   53     | 26 | 19 |
> | MemReasoner-1.4B (bAbi) |   83 | 58 | 50 |
>
>
> [1] Sukhbaatar, Sainbayar, Jason Weston, and Rob Fergus. "End-to-end memory networks." Advances in neural information processing systems 28 (2015).

---

### Official Review · Reviewer_vesZ · 2024-11-07

**Soundness:** 3
**Presentation:** 2
**Contribution:** 2
**Rating:** 3
**Confidence:** 3

**Summary:**

MemReasoner is a novel encoder component of an encoder-decoder transformer architecture. On top of a transformer encoder stack, it uses a recurrent component to write memory and read memory.

The readout is fed into the decoder similar to key-value cache to improve performance on question-answeringe tasks.

The paper benchmarks their method on BABILong. Their method solves the task as shorter contexts but does not seem to length generalize. However, their method does do better than the baselines when symbolically manipulating locations.

**Strengths:**

The hypothesis and method are well motivated. The experiments are well done and the model well in short contexts and is robust to symbolic manipulation of the locations.

**Weaknesses:**

I don't think I entirely understand the comparisons. The model does much worse than the models that are finetuned on long context data. We should see if MemReasoner also benefits from this?

**Questions:**

The encoder and memory components have some cost and overhead. What are they?

How does it compare to just give the decoder more parameters?

Can MemReasoner augment existing decoder-only LLMs?

---

> ### Author Response · Authors · 2024-11-25
>
> We thank the reviewer for their feedback.  Please see our responses below:
>
> > The model does much worse than the models that are finetuned on long context data. We should see if MemReasoner also benefits from this?
>
> We thank the reviewer for their feedback. We want to clarify the misunderstanding/misinterpretation of the statement "Their method solves the task as shorter contexts but does not seem to length generalize." We understand that the reviewer here is referring to RMT and Mamba baselines in Tables 1 and 2 of the paper, that are finetuned on BABIlong samples. However,  the goal of this work is to propose a LLM architecture that robustly learns the core reasoning task and does not get confused when evaluated on long context test samples that contain naturally occuring text as distractors together with the core reasoning facts. Therefore, contaminating the training data of MemReasoner with BABIlong samples will defeat the purpose. That is why we train the proposed MemReasoner model on original Babi samples only and test them on BABIlong data, rather than training the model on BABIlong samples directly, as that will beat the purpose of testing model's reasoning ability on an unseen test distribution while the core reasoning task to be solved remains the same. Further, training on longer sequences is time and cost-intensive.  Please refer to global response section on "Finetuning on long data". The results shown in Tables 2 and 3, combined with the new results on fine-tuned Qwen models, clearly establish this point that current transformer-based LLMs (including those that support long context) as well as alternative architectures like recurrent memory transformer (RMT) and Mamba, when finetuned on Babi, does not generalize to BABIlong. Whereas MemReasoner outperforms those baslines on both single and two-hop tasks. We further show MemReasoner's robustness to manipulations of answer locations in test samples for both single and two-hop tasks, as well as on generalzation of two-hop Babi-finetuned model on single-hop BABIlong samples. We have also now added a new task of variable tracking (both in single-hop and two-hop setting), where MemReasoner again outperforms RMT and Mamba baselines.
>
>
>
> > The encoder and memory components have some cost and overhead. What are they?
>
> Please refer to global response section on "Computational cost of MemReasoner".
>
> > How does it compare to just give the decoder more parameters?
>
> Please refer to global response section on "Comparison to decoder-only LMs that support long contexts".
>
> > Can MemReasoner augment existing decoder-only LLMs?
>
> MemReasoner is a model-agnostic way to augment current decoder-only llms with a dynamically updatable memory. The MemReasoner architecture adds an encoder, a memory module, and a temporal encoding module on top of the decoder. The encodings from the encoder are used by the decoder via the past key values of the decoder layers.  Via end-to-end training, the architecture learns to write the latent encodings in a fixed-size memory, order them in their order of apperance in the context, and perform multiple hop over that context and update the latent query accordingly. The decoder learns a diffrentiated attention mechanism to the final readout from the memory, in order to accurately generate the final answer and supporting facts (intermediate hops). This ability allows MemReasoner to perform robust multi-hop reasoning on unseen test samples.
>
> Our provided experimental results in the main paper can be thought of as augmenting GPT2-L. To further demonstrate the general applicability of MemReasoner as a way to augment another existing decoder-only LLM of a different scale, we augmented GPT-J-6B with MemReasoner-like architecture and training, and showed the results in Table R1.
>
> Table R1 (Performance on BABILong task 1 and 2 using MemReasoner-6B (bAbi) with GPT-J decoder):
> | Model | 0k | 1k | 2k | 4k | 8k | 16k | 32k | 64k | 128k |
> | -------- | --------     | -------- | -------- | -------- | -------- | -------- | -------- | -------- | -------- |
> | Task 1    |  98 | 82 | 77 | 65 | 60 | 68 | 70 | 65 | 67
> | Task 2     |   98 | 65 | 50 | 34 | 35 | 32 | 22 | 27 | 30

---

### Author Response · Authors · 2024-11-22
**Global Response**

We thank the reviewers for their feedback.  We are encouraged that the reviewers find that the problem and the proposed method of length generalization for multi-hop reasoning are well motivated [vesZ, PgX8], and that our experimental results support that MemReasoner has a robust ability to generalize to long contexts while only trained on short contexts [N6d6, PgX8] and is robust to symbolic manipulation of the locations names [vesZ].  We now address some common questions from reviewers below and will be adding these points into the updated manuscript:

**Computational cost of MemReasoner [vesZ, N6d6, PgX8]**
We provide the inference cost measured in seconds per input for evaluating with Babilong in comparison to the base decoder (gpt2-large). We note that gpt2-large does not support context lengths longer than 1024 tokens.  Overall, we observe that the increase in inference time for MemReasoner is very small for 0k and MemReasoner is more efficient for 1k context length. This is because of utilizing the latent encodings of context, performing one-shot write to the memory, and executing multiple-hops over that memory in latent space.

|  Babilong | 0k | 1k | 2k | 4k | 8k | 16k | 32k | 64k | 128k |
| -------- | -------- | -------- |-------- | -------- |-------- | -------- |-------- | -------- |-------- |
| gpt2-large | 0.28 | 1.13 | - |- |- |- |- |- |-
| MemReasoner     |  0.30 | 0.33 | 0.40 | 0.61 | 0.98 | 1.94 | 3.26 | 11.25 | 13.77 |

**Finetuning on long data [vesZ, PgX8]**

We want to clarify that the goal of this work is to propose an LLM-based architecture that learns to reason over long-context in an efficient, robust, and generalizable manner. As such, the evaluation framework corresponds to a set-up where the core reasoning facts are diluted in the presence of irrelevant natural text distractors distributed over the context. This setup allows one to test how consistently language model can solve the same reasoning task across different input lengths. This is inspired by the recent research showing that current LLMs’ reasoning performance degrade at much shorter input lengths than their technical maximum (Levy, et al. https://arxiv.org/abs/2402.14848v1, Kuratov, et al. https://arxiv.org/abs/2406.10149). As a solution, we propose MemReasoner, which shows robust reasoning on unseen test distribution with longer length and/or distinct characteristics. At the same time, finetuning on longer sequences presents several practical challenges: (1) The longer sequences with proper (human or machine) annotation should be available during training -- which is typically expensive and is difficult to scale in real-world; (2) Expanding the context window usually incurs a quadratic increase in computational and memory cost for transformer-based LLMs. For example, the  training setup used in (Fu et al. https://arxiv.org/abs/2402.10171) shows that extending the Llama-2 7B model’s context window from 4k to 80k requires 8 A100 GPUs (80G each) for five days. The costs of resources and time further increase significantly for larger models, for longer context length, and for more extended training periods. (3) The test distribution is still expected to match to longer sequences seen during training -- which is not realistic. As a result, straightforward continual pre-raining or fine-tuning on longer sequences may still not solve the fundamental problem of learning to reason over (long) context in a robust and generalizable manner.

---

> ### Author Response · Authors · 2024-11-22
> **Global Response pt 2**
>
> **Additional datasets [N6d6, PgX8]**
>
> We agree that it is important to show generalization on another dataset and provide experimental results for MemReasoner for the variable tracking (VT) from RULER [1].  In the VT task, the model is given context with lines with information about variable value assignment such as "VAR AAAAA = 16438" or "VAR BBBBB = AAAAA" and the model is prompted to obtain all variables with a specific value. Variable names have the format of 5 repeating letters randomly sampled from the alphabet.  We train and evaluate with chains of length 2, 4, 6, 8 , and 10 and return the average accuracy over all chain lengths for the 1 hop and 2 hop VT tasks.  In order to pad the context for lengths 1k, 4k, and 16k, we follow the approach taken from RULER of padding with the sentence "The grass is green. The sky is blue. The sun is yellow. Here we go. There and back again.\n" until the context reaches the desired length.  This noise is not present during training and the 0k data follow the same distribution as the training data.
>
> Since VT asks for all variables with a specific value, for MemReasoner, we take all unordered readouts of the model and pass them individually to the decoder to get the variables from each reasoning hop, and then concatenate these variables in order to obtain the final answer.  For RMT, we train with 2 segments, with segment size set to the median length on the train dataset.  We observed that it is difficult to train RMT with 2 segments for the 2-hop VT task, RMT can easily learn a shortcut and have high accuracy on the training set, but does not generalize well to the test set at 0k length and performance degrades further at longer context length.
>
> Single hop:
> | Model type  | 0k | 1k | 4k | 16k
> | -------- | -------- | -------- |-------- |-------- |
> | RMT-.77B (VT) | 100 | 5.7 | 5.0 | 4.5
> | MemReasoner-1.4B (VT) | 99.9 | 100.0 | 99.9 | 99.9
>
>
> Two hop:
> | Model type  | 0k | 1k | 4k | 16k
> | -------- | -------- | -------- |-------- |-------- |
> | RMT-.77B (VT) | 74.6 | 5.7 | 1.3 | 0.2
> | MemReasoner-1.4B (VT) | 98.4 | 97.6 | 97.0 | 98.0
>
> [1] Hsieh, C. P., Sun, S., Kriman, S., Acharya, S., Rekesh, D., Jia, F., ... & Ginsburg, B. (2024). RULER: What's the Real Context Size of Your Long-Context Language Models?. arXiv preprint arXiv:2404.06654.
>
> **Comparison to decoder-only LMs that support long contexts [vesZ, PgX8]**
>
> We experiment with [Qwen2.5-0.5B](https://huggingface.co/Qwen/Qwen2.5-0.5B) and [Qwen2.5-1.5B](https://huggingface.co/Qwen/Qwen2.5-1.5B) models both of which support long context windows (up to 128k tokens) and provide results that are to be compared with those in the respective Tables 1-5 in the manuscript.  Overall, we find that MemReasoner is able to achieve better length generalization to long contexts compared to Qwen2.5.  For location changes, we find that MemReasoner outperforms both Qwen2.5-0.5B and Qwen2.5-1.5B for Task 1, but for Task 2 Qwen2.5-1.5B outperforms MemReasoner.
>
> Table 1 (Performance on bAbi tasks):
> | Model | Task 1 | Task 2|
> | -------- | -------- | --------|
> | Qwen2.5-0.5B (bAbi) |  100   | 96   |
> | Qwen2.5-1.5B (bAbi) |  99.9  | 98.9 |
> | MemReasoner-1.4B (bAbi) |    100   |    100   |
>
>
>
> Table 2 (BABILong Task 1 Results):
> | Model | 0k | 1k | 2k | 4k | 8k | 16k | 32k | 64k | 128k |
> | -------- | --------     | -------- | -------- | -------- | -------- | -------- | -------- | -------- | -------- |
> | Qwen2.5-0.5B (bAbi)     |   94   |   66   |  34  |   23   |   10    |  3   |   1    |  -    | -   |
> | Qwen2.5-1.5B (bAbi)     |   100  |   81   |  57  |   42   |   28    |  32  |   18   |  -    |  -  |
> | MemReasoner-1.4B (bAbi) |   99   |   91   |   83 |   76   |   74    | 71   | 68     | 70    | 65  |
>
>
>
> Table 3 (BABILong Task 2 Results):
> | Model | 0k | 1k | 2k | 4k | 8k | 16k | 32k | 64k | 128k |
> | -------- | -------- | -------- | -------- | -------- | -------- | -------- | -------- | -------- | -------- |
> | Qwen2.5-0.5B (bAbi) |   96   |   76   |  59  |   39   |   19    |  11  |   3    |  -    | -   |
> | Qwen2.5-1.5B (bAbi) |   99   |   67   |  32  |   25   |   10    |  6   |   2    |  -    |  -  |
> | MemReasoner-1.4B (bAbi)              |   100 |   73 |   61 |   46 |   23 | 20    | 19    | 17    | 20     |
>
>
> Table 4 (Robustness to location changes in bAbi test set):
> | Model | Task 1 | Task 2|
> | -------- | -------- | --------|
> | Qwen2.5-0.5B (bAbi) | 44.2 | 14.5  |
> | Qwen2.5-1.5B (bAbi) | 75.2 | 63.5  |
> | MemReasoner-1.4B (bAbi) |     87.2 |     52.7 |
>
>
> Table 5 (Performance on bAbi task 2 → BABILong task 1 generalization):
> | Model | 0k | 1k | 2k | 4k |
> | -------- | -------- | -------- | -------- | -------- |
> | Qwen2.5-0.5B (bAbi) |   97     |   71     |  47      |   32     |
> | Qwen2.5-1.5B (bAbi) |   100    |   58     |  36      |   18     |
> | MemReasoner-1.4B (bAbi) |   83 |   58 |   50 |   45 |

---

### Meta-Review · Area_Chair_ETmN · 2024-12-16

**Metareview:**

The paper presents a memory-augmented LLM architecture for improved temporal and multi-hop reasoning over long contexts. The key innovation is integrating an episodic memory module with temporal encoding and iterative query updates. While results show strong performance on BABILong compared to baselines (99% vs 36% accuracy for 1k context), there are significant concerns. The method requires explicit supervision for reasoning steps and predefined hop counts, limiting generalizability. The evaluation is restricted to synthetic datasets, and there's insufficient analysis of computational costs and scaling properties. The presentation lacks clarity, particularly in the methodology section and architectural diagrams. Though the core idea of memory-augmented reasoning is valuable, these limitations make the current work insufficient for acceptance.

**Additional Comments On Reviewer Discussion:**

The author response addressed several concerns but key issues remain unresolved. While they added Qwen2.5 baselines and variable tracking experiments showing MemReasoner's advantages, the fundamental concern about requiring hop-level supervision persists. Their justification for avoiding real-world datasets due to shortcuts/contamination is reasonable, but limits practical applicability. New experiments with RMT using supporting fact supervision (RMT w/ SP) showed this alone doesn't enable generalization, supporting their architectural choices. However, the authors' choice to avoid training on longer sequences to maintain "pure" evaluation conditions makes comparisons with baselines that benefit from length exposure potentially unfair. After discussion, initial reviewer scores remained unchanged (3,6,3), supporting rejection despite interesting core contributions.

---

### Decision · Program_Chairs · 2025-01-22

Reject